# Prototypical Information Bottlenecking and Disentangling for Multimodal Cancer Survival Prediction

**Yilan Zhang** [†21], **Yingxue Xu** [†1], **Jianqi Chen** [2], **Fengying Xie** [*2], **Hao Chen** [*1]

[1]The Hong Kong University of Science and Technology, [2]Beihang University
`yxueb@connect.ust.hk, jhc@cse.ust.hk`
`{zhangyilan, cjqchenjianqi, xfy_73}@buaa.edu.cn`

## Abstract

Multimodal learning significantly benefits cancer survival prediction, especially the integration of pathological images and genomic data. Despite advantages of multimodal learning for cancer survival prediction, massive redundancy in multimodal data prevents it from extracting discriminative and compact information: (1) An extensive amount of intra-modal task-unrelated information blurs discriminability, especially for gigapixel whole slide images (WSIs) with many patches in pathology and thousands of pathways in genomic data, leading to an "intra-modal redundancy" issue. (2) Duplicated information among modalities dominates the representation of multimodal data, which makes modality-specific information prone to being ignored, resulting in an "inter-modal redundancy" issue. To address these, we propose a new framework, **P**rototypical **I**nformation **B**ottlenecking and **D**isentangling (PIBD), consisting of Prototypical Information Bottleneck (PIB) module for intra-modal redundancy and Prototypical Information Disentanglement (PID) module for inter-modal redundancy. Specifically, a variant of information bottleneck, PIB, is proposed to model prototypes approximating a bunch of instances for different risk levels, which can be used for selection of discriminative instances within modality. PID module decouples entangled multimodal data into compact distinct components: modality-common and modality-specific knowledge, under the guidance of the joint prototypical distribution. Extensive experiments on five cancer benchmark datasets demonstrated our superiority over other methods. The code is released [1].

## 1 Introduction

Cancer survival analysis (Cox, 1975; Jenkins, 2005; Salerno & Li, 2023) aims to estimate the death risk of patients for prognosis, in which multimodal learning by integrating both histological information and genomic molecular profiles can benefit the prognosis of a majority of cancer types (Chen et al., 2020; 2022b; 2021; Jaume et al., 2023; Xu & Chen, 2023). These modalities offer diverse perspectives for patient stratification and informing therapeutic decision-making (Zuo et al., 2022). For example, histological images give visual phenotypic information about tumor microenvironment, e.g., the organization of cells (Jackson et al., 2020), for different grading of cancer, while genomics data provides global landscapes (Győrffy, 2021) for various molecular subtyping of cancer. They collaboratively contribute to different survival outcomes. Nevertheless, a large quantity of redundancy in mulitmodal data poses some significant challenges to effective fusion.

The primary question at hand is: *How can we capture the discriminative information from single modality by eliminating its redundancy, referred as "intra-modal redundancy" issue?* The label for a WSI consisting of numerous patches is typically provided at the WSI level, leading to weak supervision for survival prediction. In the absence of precise annotations, such as patch-wise labeling for cancerous regions in WSIs, both task-related and irrelevant information become intermingled in the model's input, resulting in information redundancy (Hosseini et al., 2023). Specifically, the region of interest, e.g., the tumor cells highly related to risk assessment, only occupies a small

---

∗ Corresponding authors, † Equal contribution.
[1]`https://github.com/zylbuaa/PIBD.git`

portion of gigapixel WSIs with high resolutions of about $100,000 \times 100,000$ pixels (Zhu et al., 2017). For this fine-grained visual recognition, although certain multiple-instance learning (MIL) (Ilse et al., 2018; Li et al., 2021; Yao et al., 2020) have provided some promising solutions, they do not enforce constraints to remove redundant information, thus struggling to obtain discriminative representations. A similar redundancy issue emerges in genomic modality. Research (Jaume et al., 2023; Chen et al., 2021) indicates that biological pathway-based gene groups, characterized by known interactions in unique cellular functions, offer more semantic correspondence with pathology features. However, these pathways can yield hundreds to thousands of groups, and only a few specific pathways exhibit a strong correlation with patient prognosis (e.g. immune-related pathways are significant for bladder cancer prognosis prediction (Jiang et al., 2021a)).

Another concern is: *How can we capture compact yet comprehensive knowledge from the dominant overlapping information in multimodal data, referred as "inter-modal redundancy" issue?* The redundancy stemming from this duplicated information can complicate the knowledge extraction. Therefore, extracting independent factors by disentangling can enhance the feature effectiveness while discarding superfluous information. The knowledge (Liang et al., 2023) can be split into distinct components: modality-specific knowledge and modality-common knowledge. The former contains information unique to a single modality, while the latter encapsulates common information and exhibits consistency across modalities. To obtain effective knowledge from multimodal redundancy, existing efforts (Chen et al., 2021; Xu & Chen, 2023) focus on integrating common information, emphasizing the inherent consistency through alignment. However, common information often dominates aligning and integrating multimodal information, leading to the suppression of modality-specific information, thereby disregarding the wealth of distinctive perspectives.

In this work, we propose a new multimodal survival prediction framework, **P**rototypical **I**nformation **B**ottlenecking and **D**isentangling (PIBD), consisting of Prototypical Information Bottleneck (PIB) module for "intra-modal redundancy" and Prototypical Information Disentanglement (PID) module for "inter-modal redundancy". First, Information Bottleneck (IB) provides a promising solution to compress unnecessary redundancy from itself while maximizing discriminative information about task targets. However, IB may suffer from the high-dimensional computational challenges posed by massive patches of a gigapixel WSI and hundreds of pathways. Instead, we propose a new IB variant, PIB, that models prototypes approximating a bunch of instances (e.g., patches of pathology or pathways of genomics) for different risk levels, which can guide selection of discriminative instances within a modality. Secondly, PID removes inter-modal redundancy by comprehensively decomposing entangled multimodal features into ideally independent modality-common and modality-specific knowledge. To do this, we reuse the joint prototypical distributions modeled by aforementioned PIB to guide the extraction of common knowledge. Simultaneously, we enforce the model to learn knowledge different from the joint prototypical distribution, which is considered as guidance for capturing modality-specific knowledge as well.

It is worth noting that the proposed method can be extended into more multimodal problems with modalities of bag structure. The contributions are as follows: (1) Inspired by information theory for mitigating redundancy, we propose a new multimodal cancer survival framework, **PIBD**, addressing both "intra-modal" and "inter-modal" redundancy challenges. (2) We design a new IB variant, **PIB**, that models prototypes for selecting discriminative information to reduce intra-modal redundancy, while **PID** addresses inter-modal redundancy by decoupling multimodal data into distinct components with the guidance of joint prototypical distribution. (3) Extensive experiments on five cancer benchmark datasets demonstrate the superiority of our approach over state-of-the-art methods.

## 2 RELATED WORKS

### 2.1 SURVIVAL PREDICTION FROM SINGLE MODALITY

Predicting survival risk is vital for understanding cancer progression. Recent advances in digital pathology (Evans et al., 2018) and high-throughput sequencing (Christinat & Krek, 2015) technologies have led to vibrant research in single-modal survival prediction using WSIs and genomics data, respectively. To handle gigapixel images, multiple-instance learning (MIL) defines a "bag" as a collection of multiple instances (i.e., image patches) and provides effective ways to learn global representations for WSIs. MIL methods focus on aggregations of instance-level predictions (Campanella et al., 2019; Feng & Zhou, 2017; Hou et al., 2016) or features (Ilse et al., 2018). For the former, bag predictions can be simply fused by pooling the probability values of instances. While the latter employs various strategies for getting the global features, e.g., clustering embeddings (Yao et al.,

2020), modeling patch correlations with graphs (Guan et al., 2022), assigning attention weights (Ilse et al., 2018; Li et al., 2021), and learning long-range interactions by transformers (Shao et al., 2021). Furthermore, genomics data provides crucial molecular information essential for survival prediction as well. Typically represented as $1 \times 1$ measurements, genomic features can be extracted using simple neural networks, e.g., MLP (Haykin, 1998) and SNN (Klambauer et al., 2017). Although these single-modality-based methods achieved remarkable improvements in feature extraction, they do not provide constraints on removing redundant information to capture the discriminative features.

## 2.2 SURVIVAL PREDICTION FROM MULTIPLE MODALITIES

In clinical practice, patients are usually collected with comprehensive multimodal data such as genomics (Klambauer et al., 2017), pathology (Zhu et al., 2017; Liu et al., 2022; Chen et al., 2022a), radiology (Jiang et al., 2021b; Yao et al., 2021), etc. for diagnosis and prognosis, thus learning multimodal interactions (Zhang et al., 2023) becomes an important motivation for many studies. These methods are broadly categorized into tensor-based and attention-based fusion techniques (Zhang et al., 2020). Some tensor-based fusions, like concatenation (Mobadersany et al., 2018) and weighted sum (Huang et al., 2020), are simple with few parameters. Alternatively, other tensor-based fusion uses bilinear pooling to create a joint representation space by computing the outer product of features, e.g., Kroncecker product (Wang et al., 2021), factorized bilinear pooling (Li et al., 2022). However, these methods are typically used in early or late fusion stages, making the inter-modal interactions (Chen et al., 2022b) prone to be neglected. Recently, attention-based fusion methods focus on learning cross-modal correlations through co-attention mechanisms (Chen et al., 2021; Zhou & Chen, 2023). For instance, MCAT (Chen et al., 2021) proposed a gene-guided co-attention, HMCAT (Li et al., 2023b) designed a radiology-guided co-attention, MOTCat (Xu & Chen, 2023) introduced the optimal transport (OT) to model the global structure consistency, and SurvPath (Jaume et al., 2023) utilized the cross-attention to model dense interactions between pathways and histologic patches. Although some approaches can partially achieve alleviating redundancy by alignment, they are prone to lose modality-specific information.

## 2.3 MULTIMODAL LEARNING WITH INFORMATION THEORY.

Recently, information theory has attracted increasing attention within the multimodal learning community due to its ability to provide measures for quantifying information (Dai et al., 2023; Liang et al., 2023; Hjelm et al., 2018). Specifically, approaches based on the information bottleneck (IB) principle (Tishby et al., 2000; Alemi et al., 2016) have emerged as effective strategies for compressing raw information while retaining task-relevant knowledge, which has found utility across multi-view (Federici et al., 2020; Lee & Van der Schaar, 2021) and multi-modal learning (Mai et al., 2022). Additionally, another kind of method centered on information disentanglement has been harnessed to extract targeted knowledge (Sanchez et al., 2020; Cheng et al., 2022; Chen et al., 2023), facilitating the learning of more compact representations. We introduce this direction into multimodal cancer survival analysis for the first time, and inspired by information theory for mitigating redundancy, we propose a new framework PIBD that provides an information perspective solution to address the massive redundancy issues in multimodal data.

## 3 METHOD

### 3.1 OVERALL FRAMEWORK AND PROBLEM FORMULATION

Given the $i$-th patient multimodal data including pathology data $\mathbf{x}_h^{(i)}$ and genomic data $\mathbf{x}_g^{(i)}$, we aim to predict patients' survival outcome by estimating a hazard function $f_{hazard}^{(i)}(t)$ that represents the risk probability of death at the time point $t$. Figure 1 displays the overall framework of our **PIBD**.

We start with extracting unimodal representations for pathology and genomics data. Following the common setting for pathological WSIs and genomic pathways in previous works (Chen et al., 2021; Jaume et al., 2023), we formulate $\mathbf{x}_h^{(i)}$ and $\mathbf{x}_g^{(i)}$ as the "bag" of instances based on multiple instance learning (MIL) for the $i$-th patient, denoted as $\mathbf{x}_h^{(i)} = \{x_{h,j}^{(i)} \in \mathbb{R}^d\}_{j=1}^{M_h}$ and $\mathbf{x}_g^{(i)} = \{x_{g,j}^{(i)} \in \mathbb{R}^d\}_{j=1}^{M_g}$, respectively, where $M_h$ is the patch numbers of a WSI and $Mg$ is the number of biological pathways.

To address "intra-modal redundancy," we propose Prototypical Information Bottleneck (PIB), detailed in Section 3.2, to select discriminative instances for each modality. Subsequently, to reduce "inter-modal redundancy", we propose Prototypical Information Disentanglement (PID) explained in Section 3.3. PID decomposes multimodal data into independent modality-common representation $C$ and modality-specific representations denoted as $S_h$ and $S_g$ for histological

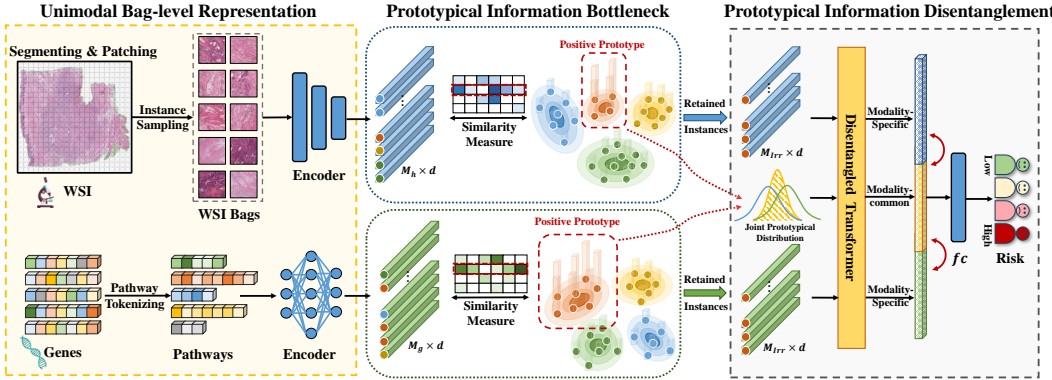

Figure 1: **Framework of PIBD**. Patient data from pathology and genomics are initially structured into bags. The Prototypical Information Bottleneck (PIB) selects discriminative features to reduce "intra-modal redundancy". Subsequently, the Prototypical Information Disentanglement (PID) module decouples the specific and common information to tackle "inter-modal redundancy".

and genomic modalities, respectively. Finally, the decoupled compact representations will be concatenated to get the final multimode features $H$, which are used for survival risk prediction.

Survival prediction estimates the risk probability of an outcome event before a specific time. However, the outcome is not always observed, resulting in right-censored data. We denote $c \in \{0, 1\}$ for censorship status ($c = 0$ means observed deaths, $c = 1$ means unknown outcomes), and discrete survival time $t \in \{1, 2, ..., N_t\}$ corresponding to a specific risk band. For a final multimodal feature $H^{(i)}$ obtained from the pathology-genomics pairs $(\mathbf{x}_h^{(i)}, \mathbf{x}_g^{(i)}, t^{(i)}, c^{(i)})$ of the $i$-th patient, we use NLL loss (Zadeh & Schmid, 2020) as survival loss function for survival prediction, following previous works (Chen et al., 2021; Xu & Chen, 2023):

$$\mathcal{L}_{surv}(\{H^{(i)}, t^{(i)}, c^i\}_{i=1}^{N_D}) = -\sum_{i=1}^{N_D} c^{(i)} log(f_{surv}^{(i)}(t|H^{(i)})) + (1 - c^{(i)}) log(f_{surv}^{(i)}(t - 1|H^{(i)}))$$
$$+ (1 - c^{(i)}) log(f_{hazard}^{(i)}(t|H^{(i)})) \tag{1}$$

where $N_D$ is the number of samples in the training sets, $f_{hazard}^{(i)}(y|H^{(i)}) = P(T = t|T \geq t, H^{(i)})$ is the hazard function representing the death probability, and $f_{surv}^{(i)}(t|H^{(i)}) = \prod_{k=1}^{t}(1 - f_{hazard}^{(i)}(k|H^{(i)}))$ is the survival function viewed as survival probability up to time point $t$. To simplify, we assume $y$ represents patient labels $(t, c)$, resulting in $2N_t$ labels.

### 3.2 PROTOTYPICAL INFORMATION BOTTLENECK

To tackle the "intra-modality redundancy", we introduce the information bottleneck and propose a new variant called Prototypical Information Bottleneck (PIB).

**Preliminary of Information Bottleneck.** The IB introduces a new representation variable $Z$ that is maximally expressive about the target $Y$, while compressing the original information from the input $X$. Thus, the objective function to be maximized is given in (Tishby et al., 2000) as:

$$R_{IB} = I(Z, Y) - \beta I(Z, X) \tag{2}$$

where $I(\cdot, \cdot)$ represents the mutual information (MI) that measures the dependence between two variables. The hyperparameter $\beta \geq 0$ acts as a Lagrange multiplier, controlling the trade-off where higher $\beta$ values lead to more compressed representations. However, the computation of MI is intractable, VIB (Alemi et al., 2016) transformed Eq.(2) into maximizing its approximation of a variational lower bound. By inverting the objective function of the variational lower bound, it tries to minimize the loss function (Derivation can be found in Appendix B.2.1.):

$$J_{IB} = \frac{1}{N} \sum_{n=1}^{N} \mathbb{E}_{z \sim p(z|x_n)}[-log q_\theta(y_n|z)] + \beta KL[p(z|x_n), r(z)] \tag{3}$$

where $N$ denotes the sample size, $q_\theta(y|z)$ is a variational approximation of the intractable likelihood $p(y|z)$, $p(z|x)$ is the posterior distribution over $z$, and $r(z)$ approximates of the prior probability $p(z)$. In practice, $r(z)$ is commonly assumed as a spherical Gaussian (Alemi et al., 2016). And the

posterior distribution $p(z|x)$ can be variationally approximated as:

$$p(z|x) \approx q_\theta(z|x) = \mathcal{N}(z; f_E^\mu(x), f_E^\Sigma(x)) \tag{4}$$

where $f_E$ is an MLP encoder that predicts both the mean $\mu$ and covariatnce matrix $\Sigma$.

**Prototypical Information Bottleneck.** IB seems to provide a hopeful solution to reduce intra-modal redundancy. However, in our task, the modality data $\mathbf{x}$ is organized as a "bag" containing numerous instances. To learn a compact bag via IB, one potential solution is to directly employ the variational approximation $q_\theta(z|x)$ of Eq.(4) in VIB to learn a representation for each instance $x \in \mathbf{x}$ in the bag. However, the drawbacks of this solution are two-fold. First, it is challenging to derive the overall distribution of the entire bag $p(z|\mathbf{x})$ for a bag $\mathbf{x}$ based on such a large number of individual instance distributions, leading to a high-dimensional computational challenge. That is, the posterior distribution $p(z|\mathbf{x})$ with respect to high-dimensional $\mathbf{x}$ of the second term in Eq.(3) is intractable. Second, since the distribution of each instance is individually learned, it is difficult to capture bag-level information for representing a compact bag. Therefore, we propose **P**rototypical **I**nformation **B**ottleneck (PIB) to directly approximate bag-level distribution $p(z|\mathbf{x})$ with a parametric distribution $p(\hat{z})$ represented by a group of prototypes, denoted as $\mathbf{P} = \{\mathcal{N}(\hat{z}; \mu_y, \Sigma_y)\}_{y=1}^{2N_t}$ (including scenarios with censored and uncensored data). To capture discriminative information about task target, each prototype is supposed to represent a conditional probability distribution $p(\hat{z}|y) = \mathcal{N}(\hat{z}; \mu_y, \Sigma_y)$ for its corresponding risk band $y$. Then, instances $z$ of a bag are expected to approach $\hat{z}$ with the same label $y$. Hence, the objective of variational approximation in Eq.(4) should become:

$$p(z|\mathbf{x}) = p(z|\mathbf{x}, y) \approx p(\hat{z}|y) \tag{5}$$

To achieve this objective, we maximize the similarity between $p(\hat{z})$ and spatial distributions of latent features $\mathbf{z} = f_E(\mathbf{x})$ for a bunch of instances, where an MLP is utilized as a representation encoder $f_E(\cdot)$ to map the input $\mathbf{x}$ to latent features $\mathbf{z}$. As a result, we just need to optimize the parametric prototypes $\hat{z}$ and $f_E(\cdot)$ for a bag $\mathbf{x}$, instead of modeling $p(z|\mathbf{x})$ for each instance of the bag.

In detail, to align the distribution of latent features $\mathbf{z}$ and parametric prototypes $\hat{z}$, we first sample some features from various prototypes via Monte Carlo sampling (to simplify the mathematical notation, we assume sampling once from each prototype). Then, we attempt to maximize the similarities between postive prototypes $\hat{z}_+$ (with true label) and the most related instances, while minimizing these instances with other negative prototypes $\hat{z}_-$. For example, given the $i$-th patient data, we have the bag features $\mathbf{z}^{(i)} = f_E(\mathbf{x}^{(i)}) = \{z_m^{(i)}\}_{m=1}^M$ and the features $\hat{\mathbf{z}}^{(i)} = \{\hat{z}_n^{(i)}\}_{n=1}^{2N_t}$ sampled from prototypes, where $M$ is the number of instances in a bag $\mathbf{x}^{(i)}$, $2N_t$ is the number of prototypes. Then, we measure the similarity between each prototype $\hat{z}_n^{(i)}$ and bag $\mathbf{z}^{(i)}$ as:

$$Sim(\hat{z}_n^{(i)}, \mathbf{z}^{(i)}) = \frac{1}{M} \sum_{m=1}^M d(\hat{z}_n^{(i)}, z_m^{(i)}) \tag{6}$$

where $d(\cdot)$ can be any similarity measure and we use cosine in our experiments. To eliminate redundant instances unrelated to risk prediction, we select a portion of instances with higher similarity scores in a bag, while the discarded instances do not contribute to the learning process. During training, since we have access to the true label, the objective of approximating $p(z|\mathbf{x}, y)$ with prototypes $p(\hat{z}|y)$ in Eq.(5) can be achieved by gathering these most related instances closer to positive $(+)$ prototypes while pushing them away from negative $(-)$ ones, formulated as:

$$\mathcal{L}_{pro} = \frac{1}{N_D} \sum_{i=1}^{N_D} -Sim(\hat{z}_+^{(i)}, \tilde{\mathbf{z}}_+^{(i)}) + \frac{1}{2N_t - 1} \sum_{n=1}^{2N_t - 1} Sim(\hat{z}_{-,n}^{(i)}, \tilde{\mathbf{z}}_{-,n}^{(i)}) \tag{7}$$

where $\tilde{\mathbf{z}}_n^{(i)} = \{z_j^{(i)} : \forall 1 \leq j \leq M_{Irr}, d(\hat{z}_n^{(i)}, z_j^{(i)}) \geq d(\hat{z}_n^{(i)}, z_{j+1}^{(i)})\}$ represents the retained instances containing task-related discriminative information with higher similarities. The retained number $M_{Irr}$ is determined by the hyperparameter $Irr$, the information retention rate (Irr), which controls the proportion of redundancy removal achieved by prototypes.

To review the objective of IB, we substitute the prototypes $\hat{Z}$ into the IB objective function in Eq.(2). After getting the approximation $p(\hat{z}|y)$ for $p(z|\mathbf{x}, y)$ or $p(z|\mathbf{x})$ in Eq.(5), we can conduct a similar derivation like from Eq.(2) to Eq.(3) (Details can be found in Appendix B.2.2), to obtain the objective loss function of PIB to be minimized as follows:

$$J_{PIB} = \frac{1}{2N_t} \sum_{n=1}^{2N_t} \mathbb{E}_{\hat{z} \sim p(\hat{z}|y_n)} [-logq_\theta(y_n|\hat{z})] + \beta KL[p(\hat{z}|y_n), r(z)] \tag{8}$$

where the first term is a cross-entropy loss for learning discriminative features. Since we are dealing with a survival prediction task with labels containing survival time and censoring status, we use the

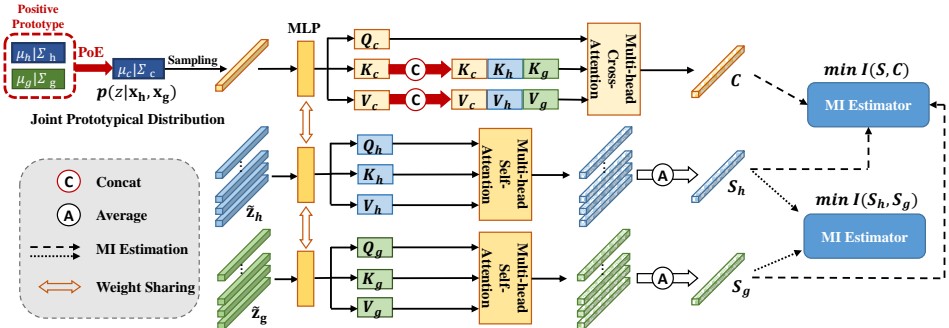

Figure 2: **Disentangled Transformer**. The self-attention is employed to model the intra-modal interactions while a token sampled from the joint prototypical distribution is used to guide common information extraction through cross-attention.

task-loss NLL in Eq.(1) as an alternative for the first term. Finally, combining the approximation term $\mathcal{L}_{pro}$, we obtain the total loss function for PIB to be minimized as follows:

$$\mathcal{L}_{PIB} = \frac{1}{2N_t} \sum_{n=1}^{2N_t} \{\alpha \mathcal{L}_{surv}(\hat{z}^{(n)}, t^{(n)}, c^{(n)}) + \beta KL[\mathcal{N}(\hat{z}; \mu_n, \Sigma_n), r(z)]\} + \gamma \mathcal{L}_{pro} \qquad (9)$$

where $\mathcal{N}(\hat{z}; \mu_n, \Sigma_n) = p(\hat{z}|y_n)$, $\alpha, \beta, \gamma$ are the hyperparameters which control the impact of items. As a result, the modeled PIB can guide the extraction of discriminative features and the removal of redundant information for each modality organized as bags.

### 3.3 PROTOTYPICAL INFORMATION DISENTANGLEMENT

After eliminating redundancy from unimodal sources, we propose a **P**rototypical **I**nformation **D**isentanglement (PID) module to decouple the shared and specific representations, addressing the "inter-modal redundancy". Suppose the instances selected by PIB are $\tilde{\mathbf{z}}_h^{(i)}$ and $\tilde{\mathbf{z}}_g^{(i)}$, we hope to decompose entangled multimodal data into ideally independent modality-common features $C^{(i)}$ and modality-specific features $S_h^{(i)}, S_g^{(i)}$. To achieve this, we reuse the joint prototypical distributions modeled by PIB for extracting common knowledge. These common features can be further used as guidance for learning modality-specific knowledge by enforcing specific knowledge independent from these shared features. Thus, we minimize the mutual information (MI) between common and specific factors to preserve modality-specific information. Consequently, our objective is to ensure the independence of specific representations within each modality and also the independence between common and specific features. The loss function of PID can be formally expressed as:

$$\mathcal{L}_{PID} = I(S, C) + I(S_h, S_g), where \ S = Cat(S_h, S_g) \qquad (10)$$

where, $S$ denotes all specific representations obtained by concatenating $Cat(\cdot)$ the features $S_h, S_g$ from each modality. As MI is intractable, we introduce an upper bound CLUB (Cheng et al., 2020) to accomplish MI minimization in Eq.(10) (Details about CLUB can be found in Appendix B.3).

To implement the above loss, we design a disentangled layer called disentangled transformer shown in Figure 2. This transformer models various interactions within the inputs thereby obtaining the features $S_h, S_g$ and $C$ required in Eq.(10). We initially extract the common information guided by the joint prototypical distribution, denoted as the joint posterior distribution $p = (z|\mathbf{x_h}, \mathbf{x_g})$, which is defined by the product-of-experts (PoE) (Cao & Fleet, 2014), an idea of combining several distributions ("experts") by multiplying them. Since we have previously obtained the positive prototype in PIB, which approximates the distribution $p(z|\mathbf{x})$ of the patient's risk band, the $p = (z|\mathbf{x}_h, \mathbf{x}_g)$ can be formulated into:

$$p(z|\mathbf{x}_h, \mathbf{x}_g) \propto p(z)p(z|\mathbf{x}_h)p(z|\mathbf{x}_g)$$
$$where \quad p(z|\mathbf{x}_h) \approx \mathcal{N}(\hat{z}; \mu_h^+, \Sigma_h^+), \ p(z|\mathbf{x}_g) \approx \mathcal{N}(\hat{z}; \mu_g^+, \Sigma_g^+) \qquad (11)$$

where $p(z)$ is the prior distribution and $p(z|\mathbf{x})$ approximately equals to the distributions of the positive prototypes $\mathcal{N}(\hat{z}; \mu^+, \Sigma^+)$. We assume the prior distribution $p(z)$ is a spherical Gaussian $\mathcal{N}(z; \mu_0, \Sigma_0)$, thus it can be shown that the product of Gaussian distributions is also a Gaussian $p(z|\mathbf{x}_h, \mathbf{x}_g) = \mathcal{N}(z; \mu_c, \Sigma_c)$:

$$\Sigma_c = (\Sigma_0^{-1} + \sum_{i \in \{h,g\}} \Sigma_i^{-1})^{-1}, \mu_c = (\mu_0 \Sigma_0^{-1} + \sum_{i \in \{h,g\}} \mu_i \Sigma_i^{-1}) \Sigma_c^{-1} \qquad (12)$$

Hence, we sample from $p(z|\mathbf{x}_h, \mathbf{x}_g)$ to obtain a guiding token for shared information extraction. The modality-common representations $C$ are then extracted by the cross-attention within the disentangled transformer. Moreover, for the modality-specific information, self-attention encodes pathway-to-pathway and patch-to-patch interactions, and their mean representation becomes $S_h$, $S_g$. Thus, under the constraint of Eq. (10), we can simultaneously extract compact features that contain both specific and common information.

**Overall Loss.** The final loss of PIBD is as follows, where $\mathcal{L}^h_{PIB}$ and $\mathcal{L}^g_{PIB}$ represent the PIB loss formulated in Eq.(9) for pathology and genomics modalities, respectively:

$$\mathcal{L} = \mathcal{L}_{surv} + \mathcal{L}^h_{PIB} + \mathcal{L}^g_{PIB} + \lambda\mathcal{L}_{PID} \tag{13}$$

where $\lambda$ is the weight factor that controls the impact of loss item, as well as $\alpha$, $\beta$, $\gamma$ in Eq.(9). Note that the proposed method can be extended to more multimodal data of the bag structure.

**Inference.** The inference process differs from the training mainly in how we find the positive prototypes. During training, with known labels, we can directly obtain the joint prototypical distribution for PID. However, in inference, we need to identify the positive one from the set of prototypes. To achieve this, we first select instances with higher similarity scores calculated with all prototypes in Eq. (7). These selected instances are considered as relevant instances. Among them, the prototype with the highest proportion of relevant instances is considered positive. Hyperparameters such as the number of samples and information retention rate remain consistent with the training process.

## 4 EXPERIMENT

### 4.1 DATASET AND IMPLEMENTATION DETAILS

We conduct extensive experiments over five public cancer datasets from TCGA[2]: Breast Invasive Carcinoma (BRCA), Bladder Urothelial Carcinoma (BLCA), Colon and Rectum Adenocarcinoma (COADREAD), Stomach Adenocarcinoma (STAD), and Head and Neck Squamous Cell Carcinoma (HNSC). We follow the work (Jaume et al., 2023) to collect the biological pathways as genomics data. 5-fold cross-validation for each dataset is employed. The models are evaluated using the concordance index (C-index) (Harrell Jr et al., 1996) and its standard deviation (std) to quantify the performance of correctly ranking the predicted patient risk sores. We also visualize the Kaplan-Meier (KM) Kaplan & Meier (1958) curves that can show the survival probability of different risk groups. The details of the dataset and experimental implementation can be found in **Appendix C.1**.

### 4.2 COMPARISONS WITH STATE-OF-THE-ARTS

We compare our method with three groups of SOTA methods: (1) *Unimodal methods*. For pathways data, we adopt **MLP** (Haykin, 1998), **SNN** (Klambauer et al., 2017), and **SNNTrans** (Klambauer et al., 2017; Shao et al., 2021) as the genomic baselines. For histology, we compare with SOTA MIL methods **ABMIL** (Ilse et al., 2018), **AMISL** (Yao et al., 2020), **TransMIL** (Shao et al., 2021) and **CLAM** (Lu et al., 2021). (2) *Multimodal methods*. Four SOTA methods are compared in this group: **Porpoise** (Chen et al., 2022b), **MCAT** (Chen et al., 2021), **MOTCat** (Xu & Chen, 2023), and **SurvPath** (Jaume et al., 2023), where we adopt two late-fusion approaches including concatenation (Cat) and Kronecker product (KP) for both Porpoise and MCAT. Besides, a prediction-level combination using a CoxPH (Cox, 1972) model of risk scores from the best-performing methods of genomics and histology is also conducted. (3) *Information theory-based methods*. As our work provides an information theory perspective on multimodal cancer survival prediction, we also compare it with information theory-based methods in multi-view, multi-modal, and task-specific fine-tuning domains, including **CLAM-SB-FT** (Li et al., 2023a), **MIB** (Federici et al., 2020), **DeepIMV** (Lee & Van der Schaar, 2021), and **L-MIB** (Mai et al., 2022). Note that although CLAM-SB-FT is an IB-based method for WSIs, it is designed within a fine-tuning framework and not be studied in multimodal survival prediction.

**Comparison.** From the results in Table 1, we can observe that PIBD achieves the best overall performance across five cancer datasets. Compared with unimodal methods[†], most multimodal methods[‡] including ours show higher overall C-index, indicating that the information from both modalities gives great perspectives and contributions to survival prediction. Note that among multimodal methods, the proposed PIBD achieves superior performance in 4 out of 5 benchmarks and outperforms the second-best method by 1.6% in overall C-index, revealing the importance of addressing intra-modal and inter-modal redundancy. Then, from the comparison between IB-based

---

[2]https://portal.gdc.cancer.gov/

Table 1: C-index (mean $\pm$ std) over five cancer datasets. g. and h. refer to genomic modality and histological modality, respectively. The best results and the second-best results are highlighted in **bold** and in underline. A method marked with the subscript † falls into the unimodal group, ‡ into the multimodal group, and ⋆ into the information theory-based group.

| Model | Modality | BRCA (N=869) | BLCA (N=359) | COADREAD (N=296) | HNSC (N=392) | STAD (N=317) | Overall |
|---|---|---|---|---|---|---|---|
| †MLP | g. | $0.622 \pm 0.079$ | $0.530 \pm 0.077$ | $0.712 \pm 0.114$ | $0.520 \pm 0.064$ | $0.497 \pm 0.031$ | 0.576 |
| †SNN | g. | $0.621 \pm 0.073$ | $0.521 \pm 0.070$ | $0.711 \pm 0.162$ | $0.514 \pm 0.076$ | $0.485 \pm 0.047$ | 0.570 |
| †SNNTrans | g. | $0.679 \pm 0.053$ | $0.583 \pm 0.060$ | $0.739 \pm 0.124$ | $0.570 \pm 0.035$ | $0.547 \pm 0.041$ | 0.622 |
| †ABMIL | h. | $0.672 \pm 0.051$ | $0.624 \pm 0.059$ | $0.730 \pm 0.151$ | $0.624 \pm 0.042$ | $0.636 \pm 0.043$ | 0.657 |
| †AMISL | h. | $0.681 \pm 0.036$ | $0.627 \pm 0.032$ | $0.710 \pm 0.091$ | $0.607 \pm 0.048$ | $0.553 \pm 0.012$ | 0.636 |
| †TransMIL | h. | $0.663 \pm 0.053$ | $0.617 \pm 0.045$ | $0.747 \pm 0.151$ | $0.619 \pm 0.062$ | $0.660 \pm 0.072$ | 0.661 |
| †CLAM-SB | h. | $0.675 \pm 0.074$ | $0.643 \pm 0.044$ | $0.717 \pm 0.172$ | $0.630 \pm 0.048$ | $0.616 \pm 0.078$ | 0.656 |
| †CLAM-MB | h. | $0.696 \pm 0.098$ | $0.623 \pm 0.045$ | $0.721 \pm 0.159$ | $0.620 \pm 0.034$ | $0.648 \pm 0.050$ | 0.662 |
| ‡SNNTrans+CLAM-MB | g.+h. | $0.699 \pm 0.064$ | $0.625 \pm 0.060$ | $0.716 \pm 0.160$ | $0.638 \pm 0.066$ | $0.629 \pm 0.065$ | 0.661 |
| ‡Porpoise(Cat) | g.+h. | $0.668 \pm 0.070$ | $0.617 \pm 0.056$ | $0.738 \pm 0.151$ | $0.614 \pm 0.058$ | $0.660 \pm 0.106$ | 0.660 |
| ‡Porpoise(KP) | g.+h. | $0.691 \pm 0.038$ | $0.619 \pm 0.055$ | $0.721 \pm 0.157$ | $0.630 \pm 0.040$ | $0.661 \pm 0.085$ | 0.664 |
| ‡MCAT(Cat) | g.+h. | $0.685 \pm 0.109$ | $0.640 \pm 0.076$ | $0.724 \pm 0.137$ | $0.564 \pm 0.840$ | $0.625 \pm 0.118$ | 0.647 |
| ‡MCAT(KP) | g.+h. | $\underline{0.727 \pm 0.027}$ | $0.644 \pm 0.062$ | $0.709 \pm 0.162$ | $0.618 \pm 0.093$ | $0.643 \pm 0.075$ | 0.668 |
| ‡MOTCat | g.+h. | $\underline{0.727 \pm 0.027}$ | $0.659 \pm 0.069$ | $0.742 \pm 0.124$ | $\mathbf{0.656 \pm 0.041}$ | $0.621 \pm 0.065$ | 0.681 |
| ‡SurvPath | g.+h. | $0.724 \pm 0.094$ | $0.660 \pm 0.054$ | $\underline{0.758 \pm 0.143}$ | $0.606 \pm 0.080$ | $\underline{0.667 \pm 0.035}$ | $\underline{0.683}$ |
| ⋆CLAM-SB-FT | h. | $0.606 \pm 0.110$ | $0.633 \pm 0.065$ | $0.725 \pm 0.150$ | $0.620 \pm 0.084$ | $0.654 \pm 0.051$ | 0.648 |
| ⋆MIB | g.+h. | $0.602 \pm 0.112$ | $0.573 \pm 0.036$ | $0.711 \pm 0.182$ | $0.555 \pm 0.055$ | $0.588 \pm 0.057$ | 0.606 |
| ⋆DeepIMV | g.+h. | $0.659 \pm 0.089$ | $0.638 \pm 0.054$ | $0.749 \pm 0.145$ | $0.604 \pm 0.061$ | $0.597 \pm 0.047$ | 0.649 |
| ⋆L-MIB | g.+h. | $0.687 \pm 0.071$ | $\underline{0.662 \pm 0.093}$ | $0.720 \pm 0.167$ | $0.615 \pm 0.085$ | $0.634 \pm 0.060$ | 0.664 |
| ⋆,‡**PIBD** | g.+h. | $\mathbf{0.736 \pm 0.072}$ | $\mathbf{0.667 \pm 0.061}$ | $\mathbf{0.768 \pm 0.124}$ | $\underline{0.640 \pm 0.039}$ | $\mathbf{0.684 \pm 0.035}$ | **0.699** |

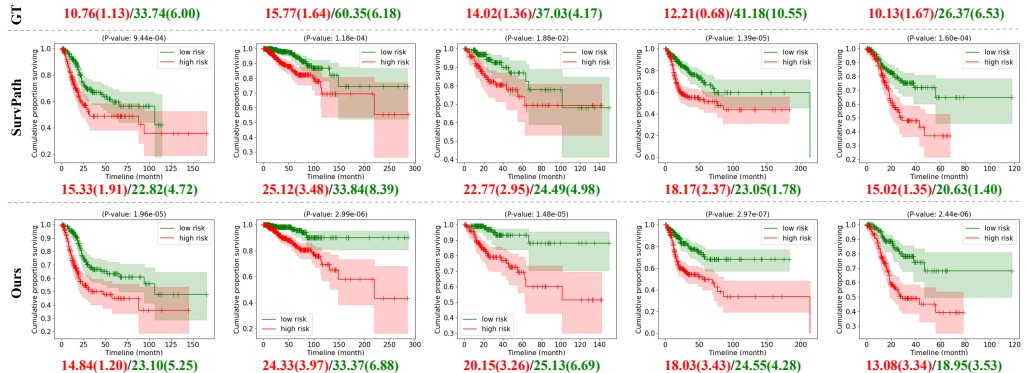

Figure 3: Kaplan-Meier curves of predicted high-risk (red) and low-risk (green) groups. A P-value < 0.05 indicates statistical significance, and the shaded regions represent the confident intervals. The median survival months are reported in the format of "high-risk: mean(std)/low-risk: mean(std)"

methods, our method achieves superior performance on all cancer datasets, with 0.5%-4.9% performance gains. PIBD, which fully considers the characteristics of bag structure under weak supervision and is designed for multimodal cancer survival prediction, demonstrates its superiority.

**Kaplan-Meier analysis** We further evaluate our method using statistical analysis, and the Kaplan-Meier curves are presented in Figure 3. Patients are separated into high-risk and low-risk groups based on predicted risk scores, with the median value of each validation set serving as the cut-off. Subsequently, we utilize the log-rank test to compute p-values, which assess the statistical significance of differences between these groups, and the median survival months are also reported for each group. Our approach demonstrates significantly improved discrimination between the two groups when compared to the second-best method, SurvPath. This effect is particularly pronounced in the BRCA, COADREAD, and HNSC datasets, with substantial margins of magnitude.

## 4.3 ABLATION STUDY

**Component validation**. In Table 2, we ablate the designs mentioned in Sections 3.2 and 3.3, which are proposed for "inter-modal redundancy" and "intra-modal redundancy". For ablating PIB, we es-

Table 2: Ablation study assessing C-index (mean $\pm$ std).

| Variants | PIB | PID | BRCA | BLCA | COADREAD | HNSC | STAD | Overall |
|---|---|---|---|---|---|---|---|---|
| AP | | | $0.684 \pm 0.044$ | $0.619 \pm 0.090$ | $0.713 \pm 0.161$ | $0.567 \pm 0.073$ | $0.609 \pm 0.048$ | 0.638 |
| PIB(AP) | ✓ | | $\underline{0.705 \pm 0.108}$ | $0.593 \pm 0.038$ | $0.753 \pm 0.143$ | $\underline{0.623 \pm 0.107}$ | $0.613 \pm 0.071$ | 0.657 |
| TransMIL | | | $0.672 \pm 0.088$ | $0.636 \pm 0.059$ | $0.750 \pm 0.133$ | $0.591 \pm 0.080$ | $\underline{0.662 \pm 0.090}$ | 0.662 |
| PIB(TransMIL) | ✓ | | $0.696 \pm 0.069$ | $\underline{0.648 \pm 0.074}$ | $\underline{0.757 \pm 0.176}$ | $0.615 \pm 0.062$ | $0.643 \pm 0.074$ | $\underline{0.672}$ |
| PIBD | ✓ | ✓ | $\mathbf{0.736 \pm 0.072}$ | $\mathbf{0.667 \pm 0.061}$ | $\mathbf{0.768 \pm 0.124}$ | $\mathbf{0.640 \pm 0.039}$ | $\mathbf{0.684 \pm 0.035}$ | $\mathbf{0.699}$ |

Table 3: Interventions in PIB. We conduct interventions by either removing the positive prototype or randomly deleting one of the negative prototypes.

| Intervention | BLCA | COADREAD | STAD |
|---|---|---|---|
| Positive | $0.401 \pm 0.086$ | $0.471 \pm 0.196$ | $0.384 \pm 0.110$ |
| Negative | $0.645 \pm 0.067$ | $0.731 \pm 0.106$ | $0.672 \pm 0.055$ |
| w/o Intervention | $0.667 \pm 0.061$ | $0.768 \pm 0.124$ | $0.684 \pm 0.035$ |

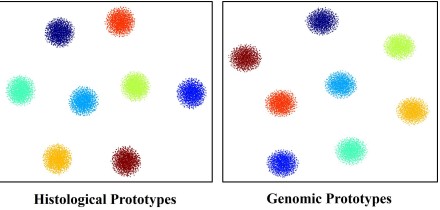

**Histological Prototypes**     **Genomic Prototypes**

Figure 4: Visualization of prototypes.

tablished two baselines: one involves direct average pooling (AP) on original features, and the other employs a non-disentangled TransMIL encoder as a strong baseline. We incorporate PIB into both baselines to assess the effectiveness of the prototypical features selected by PIB. As shown in the first four rows of Table 2, the addition of PIB outperforms the baselines in terms of higher C-index. This suggests that learning multiple distinctive prototypes in PIB and employing them to filter task-related features can effectively mitigate redundant features within each modality. For ablating PID, we conduct a comparison between our PIBD and the baseline using the non-disentangled TransMIL with PIB. The last two rows demonstrate that disentangling shared and specific information from multi-modal data effectively eliminates inter-modal redundancy, preventing the loss of modality-specific information during the fusion process and significantly enhancing the model's performance. Moreover, we conduct more quantitative studies about parameter settings presented in Appendix C.2.

**Interpretability of PIB**. To validate that the learned prototypes in PIB have modeled discriminative underlying distributions for different risk bands, we conduct random sampling on each prototype with a frequency of 2000. Subsequently, we reduce the obtained high-dimensional vectors to a two-dimensional plane using t-SNE (Van der Maaten & Hinton, 2008). As illustrated in Figure 4, the distributions exhibit excellent separability. Furthermore, inspired by the intervention in (Sarkar et al., 2022), we conduct interventions during the inference process shown in Table 3 and the results demonstrated a significant disparity. It can be seen that interventions in positive prototypes led to a dramatic decrease in the C-Index (all below 0.5), signifying a complete loss of predictive ability. Intervention in positive prototypes further results in passing a wrong guided signal to the following disentanglement module PID with an incorrect prototypical distribution as well, leading to worse performance. Conversely, when randomly removing a negative prototype, there was only a slight decline in the C-Index, which further underscores the effective modeling of discriminative risk-level distributions in PIB. Visualization of similarity scores for both modalities are presented in Appendix D.

## 5 CONCLUSION

In this work, we explore multimodal cancer survival prediction inspired by information theory and propose a new framework called PIBD aimed at addressing both "intra-model redundancy" and "inter-model redundancy" challenges. First, we propose a Prototypical Information Bottleneck (PIB) that reduces redundancy while preserving task-related information. PIB models prototypes of various risk bands, allowing us to select discriminative features from massive instances and alleviating "intra-model redundancy". Furthermore, to address "inter-modal redundancy", we propose a Prototypical Information Disentanglement (PID) to decouple independent modality-common and modality-specific features with the guidance of the joint prototypical distribution. These compact features offer distinct perspectives and knowledge, effectively enhancing the network's performance. Moreover, to handle the high-dimensional computational challenges inherent in our task, the PIB models prototypes approximating a bunch of instances by maximizing the cosine similarities within true labels. During this approximation, the choice of an appropriate similarity metric can contribute to better aligning spatial distributions, which warrants further investigation in future research endeavors.

## 6 ACKNOWLEDGMENTS

This work was supported by National Natural Science Foundation of China (No. 62202403), Shenzhen Science and Technology Innovation Committee Funding (Project No. SGDX20210823103201011), the Research Grants Council of the Hong Kong Special Administrative Region, China (Project No. R6003-22 and C4024-22GF).

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

## A  OVERVIEW

In this supplement, we will first provide detailed formulations of the proposed **PIBD** in Appendix B. Then, we provide implementation details, more quantitive studies about parameter settings, and additional comparisons in Appendix C. Finally, we display the visualization results of similarity scores for pathology and genomics modalities to further showcase our method's interpretability in Appendix D.

## B  DETAILED FORMULATIONS

### B.1  BAG FORMULATION

The bag construction process of histology and genomic data are formulated into a weakly supervised MIL task. Given a WSI, we slice it into non-overleap $M_h$ patches and use a pre-trained visual encoder to extract the features, aiming to get the patch-level $d$-dimensional embedding of each instance denoted into $\mathbf{x}_h^{(i)} = \{x_{h,j}^{(i)} \in \mathbb{R}^d\}_{j=1}^{M_h}$. For genomic data, the bag formulation process is similar. Following previous works (Jaume et al., 2023) that tokenize genes into pathways involved in particular biological processes, we use multiple separated encoders to learn the pathway-level embeddings. Then a bag of genomic data can be built as $\mathbf{x}_g^{(i)} = \{x_{g,j}^{(i)} \in \mathbb{R}^d\}_{j=1}^{M_g}$, where $Mg$ is the total number of pathways.

### B.2  DETAILED FORMULATION OF PROTOTYPICAL INFORMATION BOTTLENECK

#### B.2.1  VARIATIONAL INFORMATION BOTTLENECK (ALEMI ET AL., 2016)

The information bottleneck (IB) (Tishby et al., 2000) principle aims to find a better representation $Z$ by maximizing the mutual information (MI) between the latent representation $Z$ and label $Y$, and minimizing the MI between the $Z$ and original input $X$. The objective function to be maximized in IB can be formulated as follows:

$$R_{IB} = I(Z, Y) - \beta I(Z, X) \tag{14}$$

where $I(\cdot, \cdot)$ represents the mutual information, and $\beta$ is the Lagrange multiplier. The IB principle defines a better representation, in terms of the tradeoff between a concise representation and predictive power. However, the computation of MI is intractable, a variational information bottleneck (VIB) (Alemi et al., 2016) is proposed to give a variational approximation solution of MI estimation, which allows the use of the deep neural network for efficient training.

Given the joint distribution $p(X, Y, Z)$ as follows:

$$p(X, Y, Z) = p(Z|X, Y)p(Y|X)P(X) = p(Z|X)p(Y|X)p(X) \tag{15}$$

where assuming $p(Z|X, Y) = p(Z|X)$, corresponding to the Markov chain $Y \leftrightarrow X \leftrightarrow Z$

Then, the first part $I(Z, Y)$ can be derieved into:

$$
\begin{aligned}
I(Z, Y) &= \int dy dz p(z, y) log \frac{p(y, z)}{p(y)p(z)} \\
&= \int dx dy dz p(x)p(y|x)p(z|x) log \frac{p(y|z)}{p(y)}
\end{aligned}
\tag{16}
$$

Since $p(y|z)$ above is intractable, a decoder $q_\theta(y|z)$ is used to variational approximate the $p(y|z)$. Using the fact that the Kullback-Leibler divergence is always positive, we have:

$$
\begin{aligned}
KL[p(y|z), q_\theta(y|z)] &= \int dy p(y|z) log \frac{p(y|z)}{q_\theta(y|z)} \\
&= \int dy p(y|z) log p(y|z) - \int dy p(y|z) log q_\theta(y|z) \geq 0
\end{aligned}
\tag{17}
$$

Hence, the lower bound of $I(Z, Y)$ is as follows:

$$
\begin{aligned}
I(Z, Y) &= \int dx dy dz p(x) p(y|x) p(z|x) log \frac{p(y|z)}{p(y)} \\
&= \int dx dy dz p(x) p(y|x) p(z|x) log p(y|z) - \int dy p(y) log p(y) \\
&= \int dx dy dz p(x) p(y|x) p(z|x) log p(y|z) + H(Y) \\
&\geq \int dx dy dz p(x) p(y|x) p(z|x) log q_\theta(y|z) + H(Y)
\end{aligned}
\tag{18}
$$

where $H(Y)$ is the entropy of label $Y$, which is independent of the optimization procedure and can therefore be ignored.

Similarly, the second part $I(Z, X)$ can be formulated into its upper bound:

$$
\begin{aligned}
I(Z, X) &= \int da dx p(x, z) log \frac{p(z|x)}{p(z)} \\
&= \int dx dz p(x) p(z|x) log \frac{p(z|x) r(z)}{r(z) p(z)} \\
&= \int dx dz p(x) p(z|x) log \frac{p(z|x)}{r(z)} - KL[p(z), r(z)] \\
&\leq \int dx dz p(x) p(z|x) log \frac{p(z|x)}{r(z)}
\end{aligned}
\tag{19}
$$

where $p(z|x)$ is the posterior distribution over $z$ and $r(z)$ is a variational approximation of $p(z)$, as the computation of $p(z) = \int dx p(z|x) p(x)$ might be difficult. To sum up, the IB objective function can be seen to minimize:

$$
\begin{aligned}
J_{IB} &= - \int dx dy dz p(x) p(y|x) p(z|x) log_q \theta(y|z) + \beta \int dx dz p(x) p(z|x) log \frac{p(z|x)}{r(z)} \\
&= \frac{1}{N} \sum_{n=1}^{N} \mathbb{E}_{z \sim p(z|x_n)}[-log q_\theta(y_n|z)] + \beta KL[p(z|x_n), r(z)]
\end{aligned}
\tag{20}
$$

where $N$ denotes the number of samples. $r(z)$ can be assumed as a spherical Gaussian described in Alemi et al. (2016) in practice. And the posterior distribution $p(z|x)$ is variationally approximated by an encoder:

$$
p(z|x) \approx q_\theta(z|x) = \mathcal{N}(\boldsymbol{z}; f_E^\mu(x), f_E^\Sigma(x))
\tag{21}
$$

where $f_E$ is an MLP that predicts both the mean $\mu$ and covariance matrix $\Sigma$.

### B.2.2 PROTOTYPICAL INFORMATION BOTTLENECK

As mentioned in Section 3.2, we propose an IB variant called Prototypical Information Bottleneck (PIB), which models the prototypes approximating a bunch of instances for different risk bands.

In our task, given that the input $\mathbf{x}$ is a "bag" structure containing numerous instances, the posterior distribution $p(z|\mathbf{x})$ is intractable. Therefore, we choose to approximate $p(z|\mathbf{x})$ with a latent space distribution $p(\hat{z})$ represented by a group of prototypes $\mathbf{P} = \{\mathcal{N}(\hat{z}; \mu_y, \Sigma_y)\}_{y=1}^{2N_t}$, where each prototype represents a conditional probability distribution $p(\hat{z}|y) = \mathcal{N}(\hat{z}; \mu_y, \Sigma_y)$ under the label $y$. As a result, we just need to optimize the parametric prototypes $\hat{z}$ and $f_E(\cdot)$ for a bag $\mathbf{x}$, instead of directly employing the modeling variational approximation $q_\theta(z|x)$ of Eq.(21) in VIB to learn a compact representation for each instance of a bag.

Thus we get the loss function for the approximation:

$$
\mathcal{L}_{pro} = \frac{1}{N_D} \sum_{i=1}^{N_D} -Sim(\hat{z}_+^{(i)}, \tilde{\mathbf{z}}_+^{(i)}) + \frac{1}{2N_t - 1} \sum_{n=1}^{2N_t - 1} Sim(\hat{z}_{-,n}^{(i)}, \tilde{\mathbf{z}}_{-,n}^{(i)})
\tag{22}
$$

Afterward, we substitute the prototypes into the IB objective function in Eq.(2) and examine each item in turn. First, in our settings, since we aim to approximate $p(z|\mathbf{x}) = p(z|\mathbf{x}, y) \approx p(\hat{z}|y)$

by introducing the variable $\hat{z}$, we actually add a new chain $Y \leftrightarrow \hat{Z}$ to the original Markov chain $Y \leftrightarrow X \leftrightarrow Z$. Then, the objective function to be maximized of PIB can be formulated into:

$$R_{PIB} = I(\hat{Z}, Y) - \beta I(Z, X) \tag{23}$$

where $\hat{Z}$ is enforced to approach Z.

For the first item $I(\hat{Z}, Y)$, we got the formulation based on Eq.(18):

$$\begin{aligned} I(\hat{Z}, Y) &= \int dy d\hat{z} p(\hat{z}, y) log \frac{p(y|\hat{z})}{p(y)} \\ &\geq \int dy d\hat{z} p(\hat{z}, y) log q_\theta(y|\hat{z}) + H(Y) \end{aligned} \tag{24}$$

In our setting, hence $p(\hat{z})$ doesn't directly depend on $x$. In this case, $p(\hat{z}, y)$ can be formulated into:

$$p(\hat{z}, y) = p(\hat{z}|y)p(y) \tag{25}$$

and that we will take $p(y) = \frac{1}{2N_t}$, so we can bond the first term into:

$$\begin{aligned} I(\hat{Z}, Y) &\geq \int dy d\hat{z} p(\hat{z}, y) log q_\theta(y|\hat{z}) \\ &= \int dy dz p(y) p(\hat{z}|y) log q_\theta(y|\hat{z}) \\ &= \frac{1}{2N_t} \sum_{n=1}^{2N_t} \mathbb{E}_{\hat{z} \sim p(\hat{z}|y_n)}[log q_\theta(y_n|\hat{z})] \end{aligned} \tag{26}$$

Then for the second item $I(Z, X)$, we approximate $p(z|\mathbf{x})$ with the latent space distribution represented by IB-based prototypes:

$$p(\hat{z}) = \int dy p(y) p(\hat{z}|y) = \frac{1}{2N_t} \sum_{n=1}^{2N_t} p(\hat{z}|y_n) \tag{27}$$

Hence, we can directly replace $p(z|\mathbf{x})$ with $p(\hat{z})$ in to the second item of Eq.(20), expressed as:

$$\begin{aligned} I(Z, X) &\leq \int dx dz p(x) p(z|x) log \frac{p(z|x)}{r(z)} \\ &= \int dx dy d\hat{z} p(x) p(y) p(\hat{z}|y) log \frac{p(y)p(\hat{z}|y)}{r(z)} \\ &= \int dy p(y) log p(y) + \int dy d\hat{z} p(y) p(\hat{z}|y) log \frac{p(\hat{z}|y)}{r(z)} \\ &= -H(Y) + \frac{1}{2N_t} \sum_{n=1}^{2N_t} KL[p(\hat{z}|y_n), r(z)] \\ &\leq \frac{1}{2N_t} \sum_{n=1}^{2N_t} KL[p(\hat{z}|y_n), r(z)] \end{aligned} \tag{28}$$

So we obtain the objective loss function of PIB to be minimized as follows:

$$J_{PIB} = \frac{1}{2N_t} \sum_{n=1}^{2N_t} -\mathbb{E}_{\hat{z} \sim p(\hat{z}|y_n)}[log q_\theta(y_n|\hat{z})] + \beta KL[p(\hat{z}|y_n), r(z)] \tag{29}$$

where the first term is the task loss to learn discriminative features, we use the negative log-likelihood (NLL) loss in Eq.(1) as an alternative for the first term. Finally, after combining the approximation term $\mathcal{L}_{pro}$, we obtain the total loss function for PIB as follows:

$$\mathcal{L}_{PIB} = \frac{1}{2N_t} \sum_{n=1}^{2N_t} \{\alpha \mathcal{L}_{surv}(\hat{z}^{(n)}, t^{(n)}, c^{(n)}) + \beta KL[\mathcal{N}(\hat{z}; \mu_n, \Sigma_n), r(z)]\} + \gamma \mathcal{L}_{pro} \tag{30}$$

where $\mathcal{N}(\hat{z}; \mu_n, \Sigma_n) = p(\hat{z}|y_n)$, $\alpha$, $\beta$, $\gamma$ are the hyperparameters which control the impact of items.

### B.3 DETAILED FORMULATION OF PROTOTYPICAL INFORMATION DISENTANGLEMENT

Since in Eq.(10), the computation of mutual information (MI) is intractable, we introduce an upper bound called contrastive log-ratio upper bound (CLUB) (Cheng et al., 2020) as an MI estimator to accomplish MI minimization, which can be used in more general scenarios. Given two variables $a$ and $b$, the $I_{CLUB}(a, b)$ is calculated as follows:

$$
\begin{aligned}
I_{CLUB}(a, b) &= \mathbb{E}_{p(a,b)}[logp(b|a)] - \mathbb{E}_{p(a)}\mathbb{E}_{p(b)}[logp(b|a)] \\
&= \frac{1}{N}\sum_{i=1}^{N} logp(b_i|a_i) - \frac{1}{N^2}\sum_{i=1}^{N}\sum_{j=1}^{N} logp(b_j|a_i) \\
&= \frac{1}{N^2}\sum_{i=1}^{N}\sum_{j=1}^{N}[logp(b_i|a_i) - logp(b_j|a_i)]
\end{aligned}
\tag{31}
$$

where $logp(b_i|a_i)$ denotes the conditional log-likelihood of positive sample pair $(b_i, a_i)$ while $logp(b_j|a_i)$ is the conditional log-likelihood of negative sample pair.

However, in our work, both the modality-specific features $S_h$ and $S_g$, as well as the modality-common features $C$, are simultaneously generated by disentangled transformer, the conditional distribution $p(b|a)$ in Eq.(31) is unknown. We employ an MLP $q_\theta(b|a)$ to provide a variational approximation of $p(b|a)$, thus the variational CLUB (vCLUB), a CLUB variant, is defined:

$$
\begin{aligned}
I_{vCLUB}(a, b) &= \mathbb{E}_{p(a,b)}[logq_\theta(b|a)] - \mathbb{E}_{p(a)}\mathbb{E}_{p(b)}[logq_\theta(b|a)] \\
&= \frac{1}{N}\sum_{i=1}^{N} logq_\theta(b_i|a_i) - \frac{1}{N^2}\sum_{i=1}^{N}\sum_{j=1}^{N} logq_\theta(b_j|a_i) \\
&= \frac{1}{N^2}\sum_{i=1}^{N}\sum_{j=1}^{N}[logq_\theta(b_i|a_i) - logq_\theta(b_j|a_i)]
\end{aligned}
\tag{32}
$$

The variational approximation $q_\theta(b|a)$ can be optimized by maximizing the log-likelihood:

$$
\mathcal{L}_{estimator}(b, a) = \frac{1}{N}\sum_{i=1}^{N} logq_\theta(b_i|a_i)
\tag{33}
$$

vCLUB still holds an upper bound on MI when the variational approximation $q_\theta(b|a)$ is reliable. Therefore, to train a good estimator for the conditional distribution $q(b|a)$ is critical. In this context, we choose to predict a lower dimensional distribution conditioned on a higher dimensional distribution (Han et al., 2021) to avoid mode collapse. Following this, our work predicted $q_\theta(c|s)$ instead of $q_\theta(s|c)$ for estimating mutual information in Eq. (10) where $s$ is modality-specific distribution integrating two modalities with higher dimension, while $c$ is modality-common distribution with lower dimension. In conclusion, the proposed disentangle loss in Eq.(10) can be further defined as:

$$
\mathcal{L}_{PID} = I_{vCLUB}(S, C) + I_{vCLUB}(S_h, S_g) + \mathcal{L}_{estimator}(S, C) + \mathcal{L}_{estimator}(S_h, S_g)
\tag{34}
$$

where $S = Cat(S_h, S_g)$.

## C    More Experiments

### C.1    Implementation Details

**Dataset**. We conduct extensive experiments over five public cancer datasets from TCGA[3]: Breast Invasive Carcinoma (BRCA), Bladder Urothelial Carcinoma (BLCA), Colon and Rectum Adenocarcinoma (COADREAD), Stomach Adenocarcinoma (STAD), and Head and Neck Squamous Cell Carcinoma (HNSC). We follow the work (Jaume et al., 2023) to predict disease-specific survival (DSS), which is a more accurate representation of the patient's disease status than overall survival. For histological data, we collect all diagnosis WSIs used for primary diagnosis. For genomic data, we get the raw transcriptomics from the Xena[4] database along with DSS labels. The human biological pathways (Qu et al., 2021), represented as transcriptomics sets with specific interactions among molecules in cells, are collected from two resources by selecting where at least 90% of transcriptomic accessible: the Human Molecular Signatures Database (MSigDB) - Hallmarks (Subramanian et al., 2005; Liberzon et al., 2015) (50 pathways from 4,241 genes) and the Reactome (Gillespie et al., 2022) (281 pathways from 1577 genes).

**Evalution**. We employ 5-fold cross-validation for each dataset. The models are evaluated using the concordance index (C-index) (Harrell Jr et al., 1996) and its standard deviation (std) to quantify the performance of correctly ranking the predicted patient risk sores. We also visualize the Kaplan-Meier (KM) Kaplan & Meier (1958) curves that can show the survival probability of different risk groups. The log-rank statistical significance test Mantel et al. (1966) is performed to determine if the separation between these groups is statistically significant.

**Bag constraction**. We first segment the tissue from the images and extract non-overlapping $224 \times 224$ patches at the $20\times$ magnification. Then a Swin Transformer (CTransPath) (Wang et al., 2022), which is pre-trained on more than 14 million pan-cancer histopathology patches via self-supervised contrastive learning, is used as the feature extractor to get 768-dimensional embeddings. Meanwhile, the feature extractors of pathways are SNNs following the settings in works (Jaume et al., 2023; Xu & Chen, 2023; Chen et al., 2021).

**Implementation**. The proposed algorithm is implemented in Python with Pytorch library and runs on a PC equipped with an NVIDIA A100 GPU. For the survival prediction task, we divide the overall survival time into four intervals, resulting in eight different risk bands when considering censorship status. Therefore, we set the number of prototypes in PIB to 8. We use an MLP with a 512-d hidden layer as the latent vector encoder $f_E(\cdot)$ to embed the bag features into a fixed dimension of 256. The hyper-parameters $\alpha$, $\beta$, $\gamma$, and $\lambda$ are set to 0.1, 0.01, 1, and 0.1 respectively. We take the top 50% and 80% of the samples with the highest similarity to prototype as the retained features for histological data and genomics data, respectively, to remove redundancy within the modalities. To increase the variability during training, 4096 patches are randomly sampled from the WSI. We follow the idea in (Xu & Chen, 2023) and split the WSI bag into sub-bags, each bag has 512 instances. We use Adam (Kingma & Ba, 2014) as the optimizer with the learning rate of $5 \times 10^{-4}$. All the networks including compared methods are trained for 30 epochs and the batch size is set to 32. We report the 5-fold averaged C-index on validation sets.

### C.2    More Quantitive Studies

In this part, we reveal more experimental results about the parameter settings on three cancer datasets, shown in Figure 5. We adopt a careful approach to address the potential risk of selection bias in hyperparameter choices. To begin with, we defined the hyperparameter search space based on the commonly used ranges in information bottleneck methods like VIB (Alemi et al., 2016) and MIB (Federici et al., 2020). During the selection, we implement a grid search strategy. For each hyperparameter, while keeping others fixed, we conducted a five-fold cross-validation, selecting the hyperparameters that exhibited the best average performance across all validation sets. Furthermore, to further mitigate selection bias, these hyperparameters were selected on multiple datasets.

**Settings of Weight Factor for Loss Function**. We conduct quantitive studies about the weight factor of loss items in Eq.(13), shown in Figure 5 (a)-(d). Among these parameters, $\alpha$, $\beta$, $\gamma$ are

---

[3]https://portal.gdc.cancer.gov/
[4]https://xenabrowser.net/datapages/

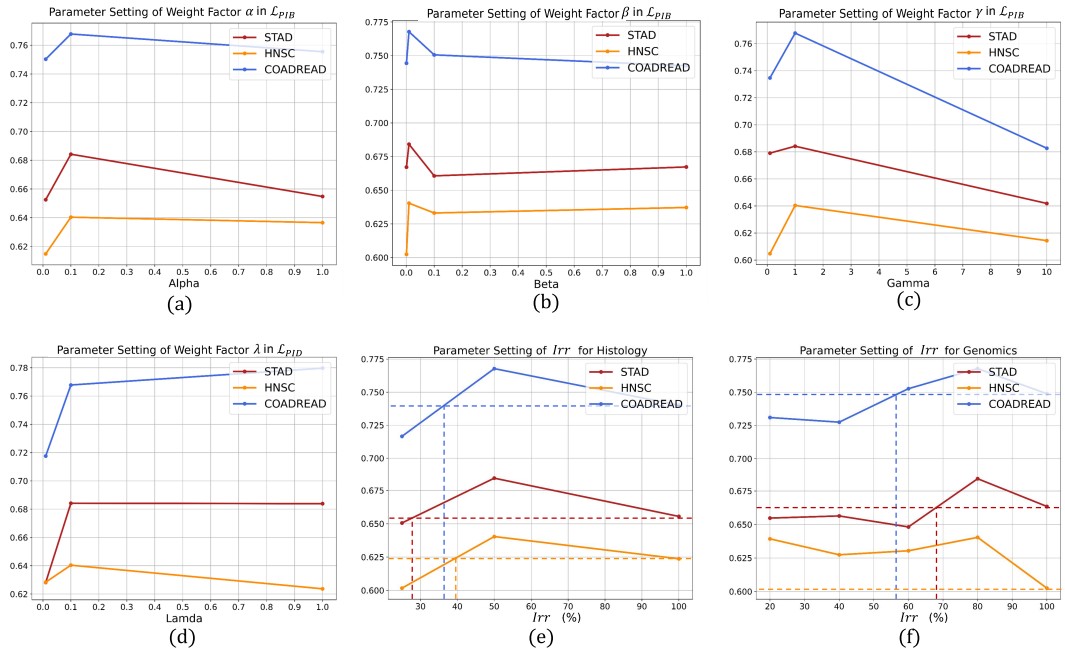

Figure 5: The effect of different parameter settings. We conduct a quantitative study on the weight factors of loss items (a)-(d) and the information retention rate of PIB (e)-(f).

for $\mathcal{L}_{PIB}$, and $\lambda$ is for $\mathcal{L}_{PID}$. In $\mathcal{L}_{PIB}$, a higher $\alpha$ implies that more label-related information is retained, a higher $\beta$ indicates more information is compressed original inputs, and a higher $\gamma$ means larger constraint on the approximation of the prototypes distributions. It can be seen that as these parameters increase, $\mathcal{L}_{PID}$ effectively removes redundant information by modeling better prototypes for various risk bands, thus enhancing the model's performance. However, when these parameters are further increased, excessive information compression within modalities may discard useful knowledge. Especially, when $\gamma$ becomes larger, the over-separability of the prototypes may reduce the generalization ability for samples deviating from the prototype's central. Then in $\mathcal{L}_{PID}$, the increase in $\lambda$ exhibits different trends across datasets. This variation could be attributed to differences in multimodal redundancy levels, which in turn affects the optimal weighting for information disentanglement. In the end, we set the $\alpha$, $\beta$, $\gamma$, and $\lambda$ to 0.1, 0.01, 1, and 0.1, respectively with better performance.

**Settings of Redundancy Removal in PIB**. $Irr$ controls the proportion of redundancy removal of uni-modalities by prototypes in PIB. A lower $Irr$ indicates that more irrelevant instances are dropped by prototypes. We can see from Figure 5 (e)-(f), when $Irr$ gradually decreases from 100% (i.e., retaining all instances), the model's performance improves, suggesting that removing intra-modal redundancy can effectively extract discriminative information. On the contrary, when $Irr$ is set too low, the model's performance deteriorates, which could indicate that task-related instances are also being discarded, resulting in the loss of valuable information. For histological data, we achieve comparable performance using only approximately 25% to 40% of the instances compared to utilizing the entire bag, resulting in a reduction in data usage of approximately 60% to 75%. Similarly, for genomic data, performance remains equivalent when utilizing approximately 55% to 70% of the dataset compared to employing all the pathways. In the end, here we set the value of $Irr$ for the pathology and genomics modalities to 50% and 80%, respectively. This allows PIB to remove redundancy while retaining effective instances, ultimately improving the model's performance.

**Settings of Sampling.** In our work, we adopt the commonly used Monte Carlo sampling method following the information bottleneck-based approaches (Alemi et al., 2016; Federici et al., 2020; Lee & Van der Schaar, 2021), leveraging the reparameterization trick and randomly sampling from a standard Gaussian distribution. Moreover, the choice of the number of samples affects the models' performances. Therefore, we gradually increase the number of samples from a small to a larger number, as illustrated in the table below. The experiments demonstrate that as the number increases,

Table 4: Settings of sample number. C-index (mean $\pm$ std).

| Sample Number | BLCA | COADREAD | STAD |
|:---:|:---:|:---:|:---:|
| 1 | $0.614 \pm 0.032$ | $0.763 \pm 0.129$ | $0.634 \pm 0.059$ |
| 10 | $0.659 \pm 0.074$ | $0.804 \pm 0.093$ | $0.673 \pm 0.082$ |
| 50 | $0.667 \pm 0.061$ | $0.768 \pm 0.124$ | $\textbf{0.684} \pm \textbf{0.035}$ |
| 100 | $\textbf{0.689} \pm \textbf{0.064}$ | $\textbf{0.811} \pm \textbf{0.128}$ | $0.678 \pm 0.060$ |

the model's performance gradually improves. Increasing the number of samples effectively aids in learning prototypes. In the paper, we strive to balance performance and sampling complexity and thus set the sample number uniformly to 50.

### C.3 MORE COMPARISONS WITH STATE-OF-THE-ARTS

To mitigate the impact of potential biases that emerge from data contamination of self-supervised models pretrained on TCGA (Guo et al., 2023; Jacovi et al., 2023), we also conduct additional experiments with a ResNet50 (He et al., 2016) encoder pretrained only on ImageNet (Deng et al., 2009) for histological modality. Here we select the top 2 well-performing methods in the unimodal group and the multimodal group in Table 1, respectively. The results are shown in Table 5. It is clearly demonstrated that even when employing a non-TCGA-pretrained encoder, PIBD exhibits great improvement compared to the comparison methods, which further underscores the superiority of our method.

Table 5: C-index (mean $\pm$ std) over five cancer datasets by using ResNet50 encoder (ImagaeNet transfer). g. and h. refer to genomic modality and histological modality, respectively. The best results and the second-best results are highlighted in **bold** and in underline. A method marked with the subscript † falls into the unimodal group, ‡ into the multimodal group

| Model | Modality | BRCA (N=869) | BLCA (N=359) | COADREAD (N=296) | HNSC (N=392) | STAD (N=317) | Overall |
|:---:|:---:|:---:|:---:|:---:|:---:|:---:|:---:|
| TransMIL [†] | h. | $0.617 \pm 0.127$ | $0.588 \pm 0.072$ | $0.699 \pm 0.170$ | $0.633 \pm 0.023$ | $0.592 \pm 0.095$ | 0.626 |
| CLAM-MB [†] | h. | $0.626 \pm 0.109$ | $0.584 \pm 0.058$ | $0.622 \pm 0.170$ | $0.606 \pm 0.031$ | $0.576 \pm 0.070$ | 0.603 |
| MOTCat [‡] | g.+h. | $0.639 \pm 0.090$ | $\underline{0.591 \pm 0.028}$ | $0.705 \pm 0.168$ | $0.589 \pm 0.067$ | $0.598 \pm 0.113$ | 0.624 |
| SurvPath [‡] | g.+h. | $\underline{0.685 \pm 0.067}$ | $0.570 \pm 0.052$ | $\underline{0.749 \pm 0.100}$ | $\textbf{0.640} \pm \textbf{0.059}$ | $\underline{0.624 \pm 0.057}$ | $\underline{0.653}$ |
| PIBD [‡] | g.+h. | $\textbf{0.697} \pm \textbf{0.092}$ | $\textbf{0.595} \pm \textbf{0.061}$ | $\textbf{0.784} \pm \textbf{0.124}$ | $\underline{0.637 \pm 0.047}$ | $\textbf{0.644} \pm \textbf{0.063}$ | **0.671** |

## D VISUALIZATION OF SIMILARITY SCORES

To show the interpretability, we have visualized similarity scores of instances to each prototype of risk level and presented top-3 relevant patches and biological pathways, as well as top-3 irrelevant biological pathways for the local interpretability of individual patients. Results from one high-risk case and one low-risk case are shown in Figure 6 and Figure 7, respectively. The visualizations clearly demonstrate that prototypes representing different risk intervals focus on distinct regions within the WSIs, enabling the extraction of discriminative instances. Conversely, instances with lower attention are considered redundant and discarded. This further elucidates how the model benefits from the redundancy removal mechanism introduced in our proposed PIB. Through analyzing these instances, we may discover biomarkers of various risk levels. This is because prototypes are designed to be discriminative, which enforces selecting instances that are most distinctive for each risk level, potentially corresponding to biomarkers for different risk levels.

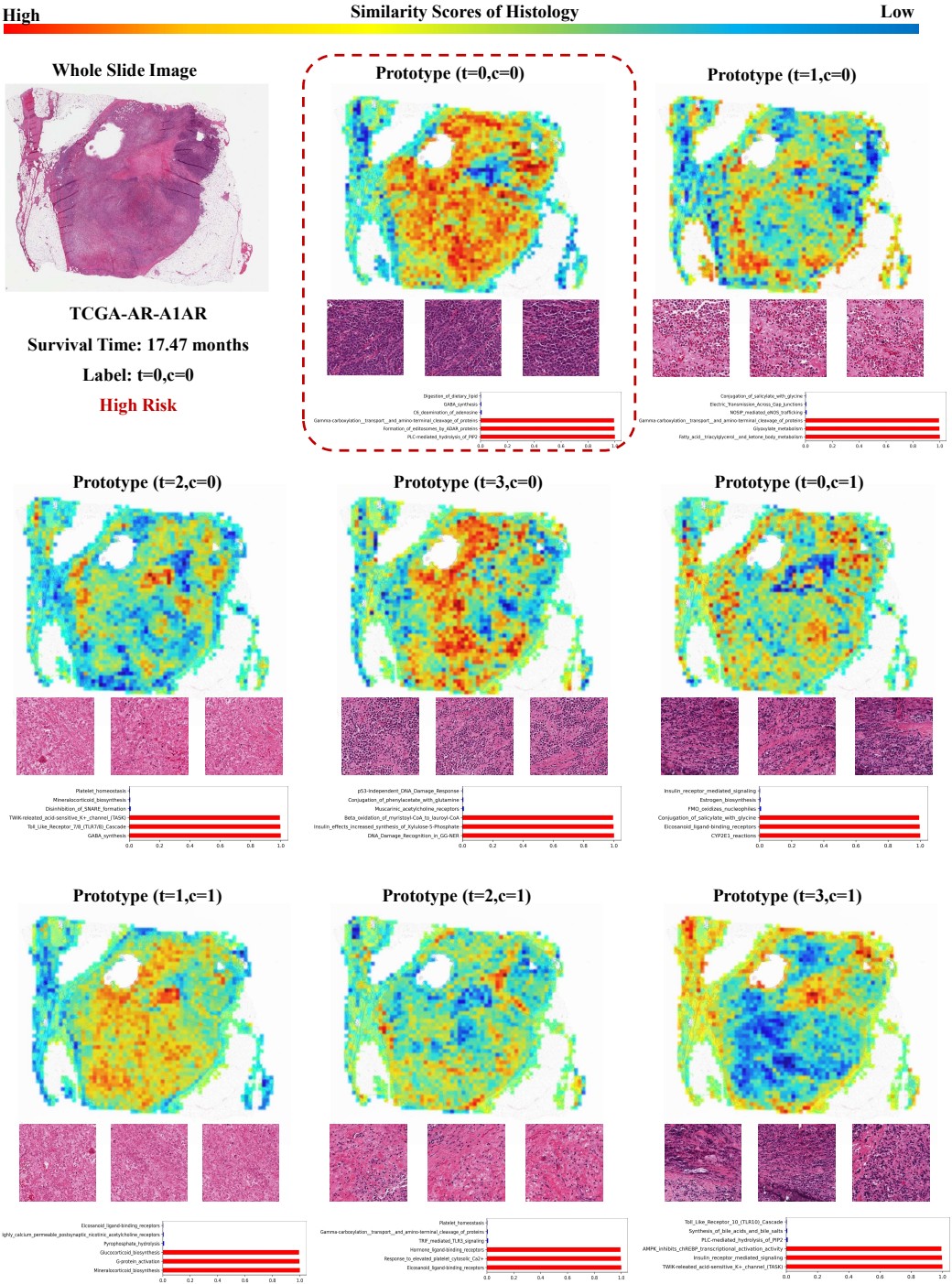

Figure 6: Visualizations of similarity scores for a high-risk case in BRCA, along with the patches displaying the highest similarities and the biological pathways showing both the lowest and highest similarities.

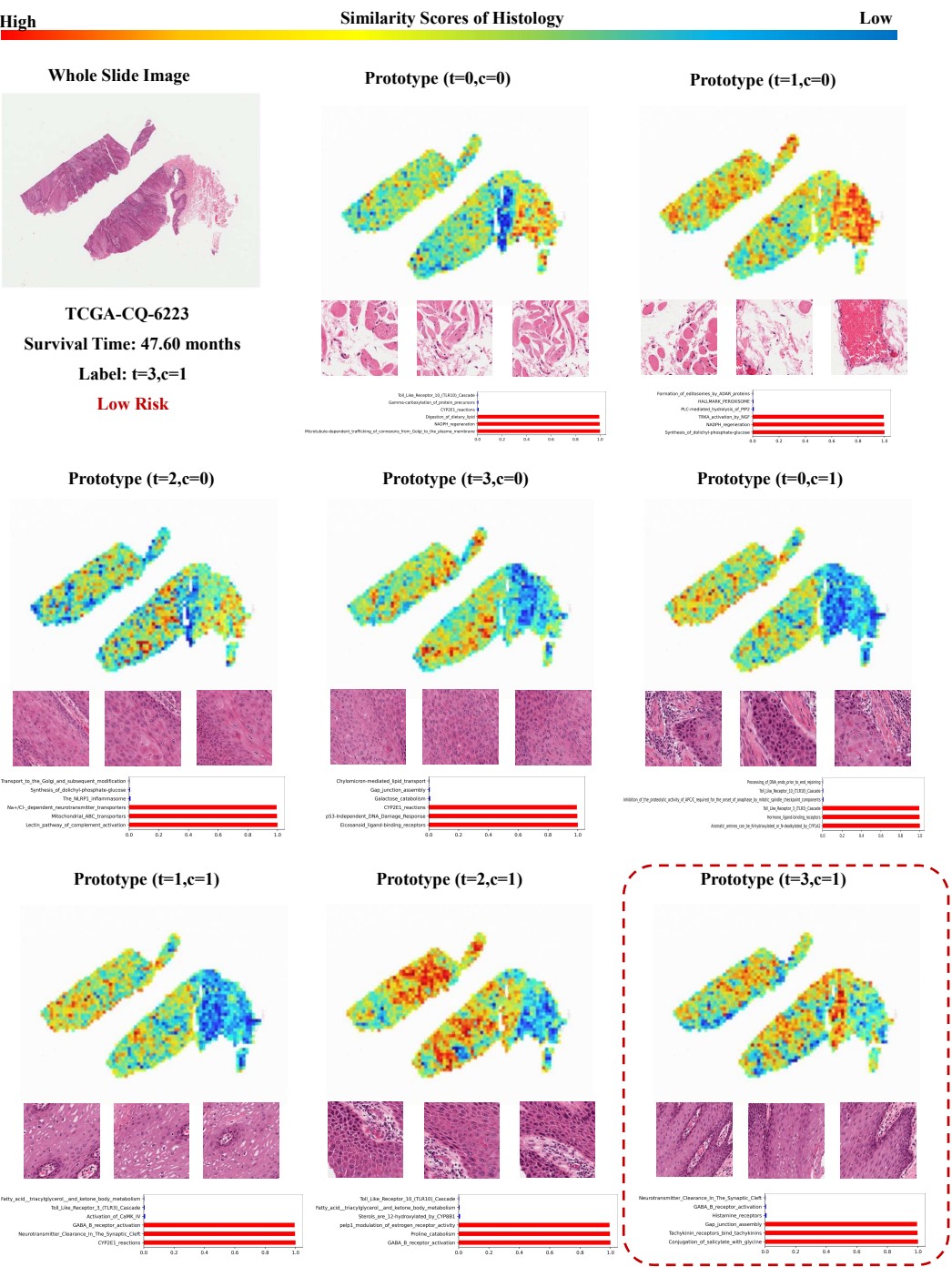

Figure 7: Visualizations of similarity scores for a low-risk case in HNSC, along with the patches displaying the highest similarities and the biological pathways showing both the lowest and highest similarities.

