# OpenReview forum: "Prototypical Information Bottlenecking and Disentangling for Multimodal Cancer Survival Prediction"
_ICLR.cc/2024/Conference — ICLR 2024 spotlight_

### Official Review · Reviewer_G86X · 2023-10-29

**Soundness:** 3 good
**Presentation:** 3 good
**Contribution:** 3 good
**Rating:** 6
**Confidence:** 4

**Summary:**

This work introduces two different information-theory-based modules for multimodal (histology and genomics) survival prediction problem. Based on information bottleneck theory, these modules aim to remove redundant information, which are prevalent in WSIs, within each modality and across different modalities. In the process, the concept of prototypes are introduced to guide the learning of the multimodal prognosis framework, while only retaining necessary information.

**Strengths:**

- While the use of information bottleneck theory is not new in computational pathology, the use of it for 1) survival prediction setting and 2) reduction of intra-modal redundant information is quite novel.
- The authors conduct extensive baseline comparisons to demonstrate that PIBD framework indeed is better/highly-competitive compared to even the most recent baselines (MOTCat and SurvPath). This is to be appreciated by any fellow researchers looking to get understanding of the available survival prediction methods in CPath.
- Probabilistic modeling of the bag of patch features (although it leads to several variational approximations), opens up new avenues for probabilistic approaches for CPath, which have been under-explored thus far.

**Weaknesses:**

Clarity
- The explanation around PIB and PID need to be much more straightforward for the work to be appreciated. In its current form, it is really hard to understand and Section 3 lacks clarity. The authors should understand that typical readership of this article will either be pathologist or ML practitioner who would be less-versed in probabilistic and variational language. This, in my opinion, significantly reduces the impact of the paper as any other researcher hoping to reuse the framework for their own experiments. Authors should try to inject more  pathology-related intuition throughout the work.

Discriminative assumption
-  I am not sure if "discriminative" approach (despite its good performance as shown in the paper) is the right approach for survival. Different from diagnosis problems where subtype/grades are indeed decided with discriminative instances (patches), survival prediction problems are not as clear-cut. Certain morphological characteristics will overlap between different risk groups (or bands), which the current approach, centered around sampling positive/negative prototypes for maximum discrimination, is not well suited for.
- Related to the comment above, this approach seems only applicable to NLL loss, which treats the problem as a classification problem and thus have "different" bins (i.e., discretized timeline). What if more common loss functions, such as Cox PH or rank losses, were to be used that do not rely on such discriminative hypothesis?

Kaplan-Meier analysis
- In Kaplan-Meier analysis section, authors state that "our approach demonstrates significantly improved discrimination.... as indicated by lower p-values". While it is tempting to compare between p-values as a metric for separation of the risk groups, it is statistically not correct. Once these are below significance threshold, it shouldn't be over-interpreted. Maybe, use median survival days for each risk group?

Sampling operation
- Since both PID and PIB rely on sampling from distribution, it does seem that the performance will indeed by affected by which samples are chosen or how many of them are sampled. The discussion around this point needs to be made explicit.

Inference
- While there are detailed information about training procedure, not much is written about the actual inference step. For instance, how many samples for each prototype are required for reliable performance?

**Questions:**

- Motivation leading up to Eq. (5) is confusing. Why is large bag size an issue, if you can learn parametric mapping (via neural network) between z and x, similar to Eq. (4)? Wouldn't this simply be a matter of backpropagation in the training, which is scalable to numerous instances?
- Variational approaches involving NNs are known to be prone to mode collapse - What are the tricks/methods used in the work to prevent it?
- I think the introduction of Markov chains confuses the paper more - Perhaps better to remove it.
- Eq. (6) - Can't we simply use KL divergence to assess the similiarity (distance), if both p(z|x) and p(\hat{z}|y) are normal distributions?
- Usually the use of "Top-k" refers to small number of instances being selected. However, from implementation section, it seems the authors are using 50~80% of the entire bag? Am I misunderstanding something?

---

> ### Author Response · Authors · 2023-11-17
> **To Reviewer G86X [1/3]**
>
> Thanks for your constructive comments and suggestions, they are exceeding helpful in improving our paper. We have carefully incorporated them in the revised paper. In the following, your comments are first stated and then followed by our point-by-point responses.
>
> ---
> > **Q1) Clarity. The clarity of the PIB and PID explanations, particularly in Section 3, needs improvement for broader appreciation. The introduction of Markov chains confuses the paper more - Perhaps better to remove it.**
> >
>
> In response to your valuable feedback, we have revised **Section 3** of the paper. The modifications include removing the introduction to Markov chains, emphasizing the changes made by the new approach more straightforward for conciseness and clarity.
>
> ---
> > **Q2) Discriminative assumption.  1. The suitability of a "discriminative" approach for survival prediction. 2. The proposed approach seems only applicable to NLL loss. What if more common loss functions, such as Cox PH or rank losses, were to be used that do not rely on such discriminative hypothesis?**
> >
> 1. We agree that survival prediction problems present greater complexity than diagnosis. In practical scenarios, there may exist deciding biomarkers within different risk bands, which leads to the “discriminative” method having demonstrated good performances in many cases [1,2].  Although some instances **related** to different risk groups may exhibit some overlap, these overlapping instances may not significantly contribute to **distinguishing** risk bands. Similarly, the same situation of overlapping can be observed in subtyping/grading as well. Therefore, our method, through the use of prototypes, excels in identifying discriminative features that are specifically associated with the positive risk level. Hence, discriminative assumption may not be a weakness but an advantage for biomarker discovery of different risk levels.
>
> 2. Indeed, the prototype learning in the current approach primarily relies on cross-entropy(CE)-like loss. This is because the prototypes are constructed based on information bottleneck theory, in which the first term of the derivation in Eq. (8) is formulated as a form of cross-entropy. Fortunately, NLL loss, a CE-like loss, makes it feasible to employ IB for survival prediction task.
>
>     Although the prototype component is based on CE-like loss, the task loss in our framework is still flexible and can be easily switched to Cox or rank loss. Furthermore, even in rank-based losses, which focus on relative comparisons between samples, discriminative information is still needed. Although not directly involving classification, the model needs to learn the differences between samples to correctly rank them. Hence, the assumption of PIB may not conflict in such scenarios.
>
>     Despite the discriminative hypothesis, this work first attempts to directly model the bag-level representations for a bag structure through the probabilistic model in survival prediction task. There is undoubtedly significant value in expanding this approach to various hypotheses like rank-based hypothesis. Hence, further investigation is still warranted in the future.
>
> **REFERENCES**
>
> [1] Echle A, Rindtorff N T, Brinker T J, et al. Deep learning in cancer pathology: a new generation of clinical biomarkers[J]. British journal of cancer, 2021, 124(4): 686-696.
>
> [2] Wiegrebe, Simon, et al. "Deep Learning for Survival Analysis: A Review." *arXiv preprint arXiv:2305.14961* (2023).

---

> > ### Author Response · Authors · 2023-11-17
> > **To Reviewer G86X [2/3]**
> >
> > > **Q3)** **Motivation. 1. Motivation leading up to Eq. (5) is confusing.  Why is large bag size an issue？if you can learn parametric mapping (via neural network) between z and x, similar to Eq. (4)? 2. Can't we simply use KL divergence to assess the similiarity (distance) in Eq(6), if both p(z|x) and p(\hat{z}|y) are normal distributions?**
> > >
> > 1. The $\bf{x}$ in Eq.(5) is organized as a “bag” structure containing numerous instances.  If we directly employ the variational approximation of Eq.(4) in VIB to learn a compact representation for each instance $x \in \bf{x}$ in the bag, every instance in a bag would have its own predicted distribution.  **The drawbacks of this solution are two-fold.** First, it is challenging to derive the overall distribution of the entire bag $p(z|\bf{x})$ for a bag $\bf{x}$ based on such a large number of individual instance distributions, leading to a high-dimensional computational challenge. That is, the posterior distribution $p(z|\bf{x})$ with respect to high-dimensional $\bf{x}$ of the second term in Eq.(3) is intractable. Second, since the distribution of each instance is individually learned, it is difficult to capture bag-level information for representing a compact bag. Therefore, we propose using prototypes to directly approximate the bag-level distribution. This simplifies the issue and allows us to capture the essence of the bag's distribution more effectively.
> > 2. From the motivation of Eq. (5), it can be inferred that obtaining $p(z|\bf{x})$ directly at the bag level is challenging. In our method, the $p(z|\bf{x})$ represents a bunch of latent features $\bf{z}$ mapped from the bag $\bf{x}$ by a representation encoder, while the prototypes $p(\hat{z}|y)$ is in a form of distribution. Thus utilizing KL divergence in Eq.(6) is not applicable. Furthermore, since we want to drop some redundant instances unrelated to the task, only some highly similar instances are selected to impose constraints In Eq.(7). Consequently, we think that computing similarity on instances is a more suitable approach.
> >
> > ---
> > > **Q4) Kaplan-Meier analysis. Comparison between p-values as a metric for separation of the risk groups is statistically not correct. Maybe, use median survival days for each risk group?**
> > >
> >
> > Thank you very much for highlighting this concern! Based on your suggestion, we have revised the relevant section on **Page 8** and incorporated median survival months as a new and more meaningful reference point.  Since we use the five-fold cross-validation, we get the median survival months for each risk group in each validation set. We report both the average median values and their standard deviation.  From the table below, it can be observed that PIBD, compared to Survpath, exhibits a more pronounced distinction in median survival months between the two risk groups across most datasets, more closely with the median values in actual grouping.
> >
> > | High-Risk Group/Low-Risk Group | HNSC | STAD | BLCA | COADREAD | BRCA |
> > | --- | --- | --- | --- | --- | --- |
> > | Ground Truth | 12.21(0.68)/41.18(10.55) | 10.13(1.67)/26.37(6.53) | 10.76(1.13)/33.74(6.00) | 14.02(1.36)/37.03(4.17) | 15.77(1.64)/60.35(6.18) |
> > | SurvPath | 18.17(2.37)/23.05(1.78) | 15.02(1.35)/20.63(1.40) | 15.33(1.91)/22.82(4.72) | 22.77(2.95)/24.49(4.98) | 25.12(3.48)/33.84(8.39) |
> > | PIBD | 18.03(3.43)/24.55(4.28) | 13.08(3.34)/18.95(3.53) | 14.84(1.20)/23.10(5.25) | 20.15(3.26)/25.13(6.69) | 24.33(3.97)/33.37(6.88) |

---

> > > ### Author Response · Authors · 2023-11-17
> > > **To Reviewer G86X [3/3]**
> > >
> > > > **Q5) Settings of sampling. Since both PID and PIB rely on sampling from distribution, it does seem that the performance will indeed by affected by which samples are chosen or how many of them are sampled. The discussion around this point needs to be made explicit**
> > > >
> > >
> > > Thanks for this very helpful comment! The discussion on sampling is indeed crucial. In the revised version, we have incorporated experimental results and discussions on this aspect into **Appendix C.2 on Page 20**.
> > >
> > > In our work, we adopt the commonly used Monte Carlo sampling method following the information bottleneck-based approaches [1-3], leveraging the reparameterization trick and randomly sampling from a standard Gaussian distribution. Moreover, the choice of the number of samples affects the models’ performances. Therefore, we gradually increase the number of samples from a small to a larger number, as illustrated in the table below. The experiments demonstrate that as the number increases, the model's performance gradually improves. Increasing the number of samples effectively aids in learning prototypes. In the paper, we strive to balance performance and sampling complexity and thus set the sample number uniformly to 50.
> > >
> > > | Sample Number | BLCA | COADREAD | STAD |
> > > | --- | --- | --- | --- |
> > > | 1 | 0.614(0.032) | 0.763(0.129) | 0.634(0.059) |
> > > | 10 | 0.659(0.074) | 0.804(0.093) | 0.673(0.082) |
> > > | 50 | $\underline{0.667(0.061)}$ | $\underline{0.768(0.124)}$ | **0.684(0.035)** |
> > > | 100 | **0.689(0.064)** | **0.811(0.128)** | $\underline{0.678(0.060)}$ |
> > >
> > > ---
> > > > **Q6) Inference. While there is detailed information about training procedure, not much is written about the actual inference step. For instance, how many samples for each prototype are required for reliable performance?**
> > > >
> > >
> > > Thank you for pointing out this issue. In the new revised version, we have included more details about the inference process on **Page 7**. About the number of samples, as discussed in response to Q5), we set it to 50.
> > >
> > > ---
> > > > **Q7) Mode collapse. Variational approaches involving NNs are known to be prone to mode collapse - What are the tricks/methods used in the work to prevent it?**
> > > >
> > >
> > > We agree that mode collapse in variational approaches is a common issue, which is investigated in many previous researches. In this work, we mainly follow the tricks used in VIB[1] and CLUB [4].
> > >
> > > - In PIB, some regularization techniques such as dropout are employed. Furthermore, the second term in Eq. (9) is essentially considered a form of regularization to the learnable distribution of prototypes by pushing it forward to a prior distribution r(z). Hence, carefully tuning the coefficient of this term is beneficial to preventing it from mode collapse.
> > > - In PID, variational approaches primarily involve mutual information estimation in Eq. (10). Here we used vCLUB estimator which is illustrated in section B.3 of appendix, where training a good predictor for the conditional distribution $p(b|a)$ of Eq. (31) is critical. In this context, one trick to avoid mode collapse is predicting a lower dimensional distribution conditioned on a higher dimensional distribution [5]. Following this, our work predicted $q_\theta(c|s)$ instead of $q_\theta(s|c)$ for estimating mutual information in Eq. (10), where $s$ is modality-specific distribution integrating two modalities with higher dimension, while $c$ is modality-common distribution with lower dimension.
> > >
> > > Regarding the details of training, we have elaborated on it in the appendix on **Page 18** to provide a reference for readers.
> > >
> > > ---
> > > > **Q8) The use of “Top-k”.**
> > > >
> > >
> > > Thank you very much for pointing out the ambiguity in our expression. To avoid confusion, we have removed the mention of using "Top-k" and replaced it with the concept of information retention rate (Irr) to represent the proportion of selected instances in the revised manuscript.
> > >
> > > **REFERENCES**
> > >
> > > [1] Alemi, Alexander A., et al. "Deep Variational Information Bottleneck." *International Conference on Learning Representations*. 2016.
> > >
> > > [2] Wan, Zhibin, et al. "Multi-view information-bottleneck representation learning." *Proceedings of the AAAI conference on artificial intelligence*. Vol. 35. No. 11. 2021.
> > >
> > > [3] Li, Honglin, et al. "Task-specific fine-tuning via variational information bottleneck for weakly-supervised pathology whole slide image classification." *Proceedings of the IEEE/CVF Conference on Computer Vision and Pattern Recognition*. 2023.
> > >
> > > [4] Cheng, Pengyu, et al. "Club: A contrastive log-ratio upper bound of mutual information." *International conference on machine learning*. PMLR, 2020.
> > >
> > > [5] Han, Wei, Hui Chen, and Soujanya Poria. "Improving Multimodal Fusion with Hierarchical Mutual Information Maximization for Multimodal Sentiment Analysis." *Proceedings of the 2021 Conference on Empirical Methods in Natural Language Processing*. 2021.

---

> > > > ### Comment · Reviewer_G86X · 2023-12-01
> > > > **Thank you for addressing the concerns**
> > > >
> > > > I thank the authors for carefully addressing my concerns. The paper looks in much better shape now and hence I am increasing my score.

---

### Official Review · Reviewer_ottr · 2023-11-01

**Soundness:** 3 good
**Presentation:** 2 fair
**Contribution:** 2 fair
**Rating:** 5
**Confidence:** 5

**Summary:**

This work presents PIBD (Prototypical Information Bottlenecking and Disentangling) for multimodal survival analysis using pathology and genomics, which extends previous progress thus far on co-attention-based early-based fusion [1] and learning information bottleneck [2] in computational pathology via two mechanisms: 1) a Prototypical Information Bottleneck (PIB) module for intra-modal redundancy and 2) a Prototypical Information Disentanglement (PID) module for inter-modal redundancy. Experimentation is performed on the same splits for disease-specific survival analysis in Jaume et al. [1], with comparisons against other relevant works (both unimodal and multimodal) and ablation experiments that assess PIB and PID independently.


References
1. Jaume, G., Vaidya, A., Chen, R., Williamson, D., Liang, P. and Mahmood, F., 2023. Modeling Dense Multimodal Interactions Between Biological Pathways and Histology for Survival Prediction. arXiv preprint arXiv:2304.06819.
2. Li, H., Zhu, C., Zhang, Y., Sun, Y., Shui, Z., Kuang, W., Zheng, S. and Yang, L., 2023. Task-specific fine-tuning via variational information bottleneck for weakly-supervised pathology whole slide image classification. In Proceedings of the IEEE/CVF Conference on Computer Vision and Pattern Recognition (pp. 7454-7463).

**Strengths:**

- As summarized above, extensive experimentation regarding comparisons, ablation studies, and assessment of multiple cancer types is performed. In particular, this work compares with many state-of-the-art multimodal methods such as MCAT (both concatenation and Kronecker Product), SurvPath, MOTCat, and other relevant information bottleneck-related works. Beyond ABMIL and TransMIL, other strong unimodal methods such as the task-specific finetuning variant of CLAM is also considered. Background information and literature is also extensive, with many important studies related to PIBD cited in a comprehensive manner.
- The proposed PID and PIB methodology, though largely adapted and inspired by other IB applications in computational pathology such as that of Li et al. 2023, is (on balance) appropriate for its extension into survival analysis w.r.t. addressing sparsity of patch features in MIL via prototypes. Figure 1 is informative and communciates how PID/PIB are used.

**Weaknesses:**

- **Limited findings w.r.t. clinical application / multimodal interpretability**: PID/PIB is intuitive for solving multimodal integration problems of WSIs and gene sets, but the demonstrated findings of this work appears limited to only improvement of c-Index performance. Methodologies such as MOTCat and SurvPath, though not approaching survival analysis from an information bottleneck perspective, similarly resolve issues regarding redundancy of patch features in MIL and reach similar findings already established in prior works. The interpretability experiments, shown as attention heatmaps in the supplement, are difficult to interpret and draw conclusions from. As the improvement in c-Index is minor, it would be valuable to investigate the interpretability of PIBD, for example: (1) performing local and global interpretability to assess individual and cohort-level image-omic drivers of disease survival, or (2) experiment+showcase unique clinical applications that would arise from PIBD that cannot be performed in other methods, which would produce new scientific findings and expand this study's significance. In particular, I found the prototypical aspect of this work to be intriguing, with concurrent works also realizing (and finding new applications) of using prototypical patterns for pathology [5].
- **Ablation Experiments for PIB/PID**: Some baselines do not seem fair in the PID/PIB ablation experiments.
- - In ablating PID, the baseline model considered was average pooling the prototypical features - a more appropriate baseline would be to use a non-disentangled Transformer like TransMIL on top of the prototypical features.
- - In ablating PIB, the reviewer is not certain what the baseline looks like (using PID without PIB), but would think that this comparison would be similar to having each modality being processed by a TransMIL-like encoder + PID (or SurvPath + PID).
- **Using CTransPath for TCGA evaluation**: A limitation of this study is the usage of CTransPath (pretrained on TCGA) for training and evaluating MIL models for survival analysis in TCGA. Understandably, access to powerful pretrained encoders for pathology is challenging, due to: (1) lack of computing power, and (2) lack of diverse and independent data outside of TCGA, with other studies in computational pathology such as Jaume et al. [1] and Filliot et al. [2] having also conducted survival analyses via TCGA-pretrained encoders. This limitation could remain as "a limitation of this study" (fitted in the conclusion) as the novelties of this work are with respect to the MIL encoder (all comparisons are performed using the same features and evaluated in the same way), in combination with the aforementioned challenges in computational pathology. However, as the broader pathology and machine learning community is becoming more aware of potential biases that emerge from data contamination of self-supervised models that are also trained using the evaluation data (as seen in LLMs and other studies [3-7]), it is important for advances going forward to recognize and resolve this issue and to develop better standardization for developing and evaluating MIL methods.

References
1. Jaume, G., Vaidya, A., Chen, R., Williamson, D., Liang, P. and Mahmood, F., 2023. Modeling Dense Multimodal Interactions Between Biological Pathways and Histology for Survival Prediction. arXiv preprint arXiv:2304.06819.
2. Filiot, A., Ghermi, R., Olivier, A., Jacob, P., Fidon, L., Mac Kain, A., Saillard, C. and Schiratti, J.B., 2023. Scaling Self-Supervised Learning for Histopathology with Masked Image Modeling. medRxiv, pp.2023-07.
3. Guo, C., Bordes, F., Vincent, P. and Chaudhuri, K., 2023. Do SSL Models Have D\'ej\a Vu? A Case of Unintended Memorization in Self-supervised Learning. arXiv preprint arXiv:2304.13850.
4. Xiang, J. and Zhang, J., 2022, September. Exploring low-rank property in multiple instance learning for whole slide image classification. In The Eleventh International Conference on Learning Representations.
5. Chen, R.J., Ding, T., Lu, M.Y., Williamson, D.F., Jaume, G., Chen, B., Zhang, A., Shao, D., Song, A.H., Shaban, M. and Williams, M., 2023. A General-Purpose Self-Supervised Model for Computational Pathology. arXiv preprint arXiv:2308.15474.
6. Jacovi, A., Caciularu, A., Goldman, O. and Goldberg, Y., 2023. Stop uploading test data in plain text: Practical strategies for mitigating data contamination by evaluation benchmarks. arXiv preprint arXiv:2305.10160.
7. Kapoor, S. and Narayanan, A., 2023. Leakage and the reproducibility crisis in machine-learning-based science. Patterns, 4(9).

**Questions:**

Primary Questions and Suggestions
- Current application and findings of PIBD overlap with other multimodal survival analysis works wr.t. to addressing patch feature redundancy, and having similar interpretability experiments. How does the contributions and findings of this work advance the field further?
- Ablation experiments for PID/PIB can have stronger baselines.
- Evaluating PIBD with a ResNet-50 encoder (ImageNet transfer) or a non-TCGA-pretrained encoder would strength the study and its findings.

Minor Questions
- How are CLAM-SB / CLAM-MB adapted for survival analysis? To the reviewer's knowledge, the implementation of the CLAM framework is mostly situated for slide classification (not survival), as the clustering constraints in CLAM are most appropriate for subtyping problems.

---

> ### Author Response · Authors · 2023-11-17
> **To Reviewer ottr [1/2]**
>
> Thanks for your constructive comments and suggestions, they are greatly beneficial in enhancing the quality of our paper. We hope our answers will address the concerns and clarify the contributions of the paper. In the following, your comments are first stated and then followed by our point-by-point responses.
>
> ---
> > **Q1) Limited findings w.r.t. clinical application / multimodal interpretability**
> >
>
> Thank you for your feedback and suggestions! In order to present the concerns in a more organized manner, we will state our response point-by-point.
>
> **Contribution**. We'd like to emphasize the contributions of our method, as highlighted by Reviewers bKh7, HNZs, and G86X. Our approach achieves innovation by reducing redundant information within and between modalities for survival prediction through the information bottleneck in a quite novel prototypical method. Methods like MOTCat and SurvPath, which utilize co-attention mechanisms, focus specifically on gene-related instances of WSIs, emphasizing modality-common information. They failed to address the issues that we’re concerned about in the following aspects. First, these approaches ignored the distinct value of modality-specific information. Second, helpful instances in a bag only occupy an extremely small portion, which means massive intra-modal redundancy. Without explicit redundancy removal, such as MOTCat and Survpath, useful information will be flooded. Lastly, they cannot directly model at the bag-level, as noted by Reviewer G86X, "probabilistic modeling on bags opens up new avenues for probabilistic approaches for CPath, which have been under-explored thus far.”
>
> **Interpretability**. For local interpretability of individual patients, we have visualized similarity scores of instances to each prototype of risk level and presented top-3 relevant patches and biological pathways, as well as top-3 irrelevant biological pathways. Through analyzing these instances, we may discover biomarkers of various risk levels. This is because prototypes are designed to be discriminative, which enforces selecting instances that are most distinctive for each risk level, potentially corresponding to biomarkers for different risk levels. In addition to what we have done, another potential way is to analyze the modality-specific features extracted from PID using CAM-based approaches, which will facilitate the discovery of unique biomarkers within each modality.
>
> Since prototypes represent the distribution for each risk band, they inherently possess a global nature. As a result, the global interpretability can be demonstrated in the following aspects. First, by collecting and organizing instances with high similarity to prototypes across a large patient cohort, we can assess cohort-level disease survival. Second, in PID, the joint distribution of histology and genomic data containing modality-common information, contributes to identifying paired instances related to both modalities and further establishing their relationship between patches and pathways. This also helps to understand their collaborative mechanisms.
>
> **Clinical applications.** We truly appreciate your suggestions and believe that the aforementioned way to interpret the model has the potential to provide some clinical value. However, the investigation of clinical applications requires collaborative efforts from experts across disciplines, such as pathologists, statisticians, AI experts, etc. For example, interpretability in clinical applications demands extensive review of pathology slides by pathologists and continuous enhancements in algorithms by AI engineers[1-3]. Therefore, it definitely takes a lot of time and effort in development and elaboration, which has gone beyond the scope of the current work. This work still primarily focuses on technical innovation in methodology. We acknowledge the importance of clinical applications, and we will continue to explore this area and hope to report the findings soon.
>
> **REFERENCES**
>
> [1] Hosseini, Mahdi S., et al. "Computational Pathology: A Survey Review and The Way Forward." *arXiv preprint arXiv:2304.05482* (2023).
>
> [2] Berbís, M. Alvaro, et al. "Computational pathology in 2030: a Delphi study forecasting the role of AI in pathology within the next decade." *EBioMedicine* 88 (2023).
>
> [3] Lu, Ming Y., et al. "AI-based pathology predicts origins for cancers of unknown primary." *Nature* 594.7861 (2021): 106-110.

---

> > ### Author Response · Authors · 2023-11-17
> > **To Reviewer ottr [2/2]**
> >
> > > **Q2) Ablation Experiments for PIB/PID. Ablation experiments for PID/PIB can have stronger baselines.**
> > >
> >
> > Thank you for your valuable feedback! First, we would like to clarify the rationale behind our ablation experiments in our original version. In our method, PID relies on the prototypes obtained from PIB to guide the disentangling of modality-common and modality-specific information by computing the joint prototypical distribution. Therefore, using PID without PIB is not feasible. To assess the effectiveness of each module, we started from a basic baseline without PIB and PID, gradually incorporating these proposed components to evaluate their individual contributions.
> >
> > However, your suggestion is highly appreciated, prompting us to reconsider the comparison using average pooling may not be strong. Following your advice, we have employed a stronger alternative, the TransMIL-encoder:
> >
> > 1. In the PID ablation experiment, as you suggested, the baseline now uses a non-disentangled Transformer, TransMIL, on top of the prototypical features. The results are shown in variant (b) of the below table.
> > 2. In the PIB ablation experiment, since using PID without PIB is not applicable, we opted to compare with a baseline using a TransMIL-like encoder on the **original features**. The performance can be found in variant (a) of the following table. This comparison allows us to evaluate the effectiveness of PIB by examining the impact of using prototypical features versus original features.
> >
> > The table below illustrates the results of the updated ablation experiments on five datasets. It is evident that, with the adoption of a stronger baseline, the effectiveness of both PIB and PID remains validated. We have incorporated these findings into the revised version **on Page 9** and provided a more in-depth analysis for better clarity.
> >
> > | Variants | PIB | PID | BRCA | BLCA | COADREAD | HNSC | STAD | Overall |
> > | --- | --- | --- | --- | --- | --- | --- | --- | --- |
> > | (a) TransMIL |  |  | 0.672(0.088) | 0.636(0.059) | 0.750(0.133) | 0.591(0.080) | $\underline{ 0.662(0.090)}$ | 0.662 |
> > | (b) PIBD (PIB+TransMIL) | √ |  |$\underline{ 0.696(0.069)}$ | $\underline{ 0.648(0.074) }$| $\underline{ 0.757(0.176) }$ | $\underline{ 0.615(0.062)}$ | 0.643(0.074) | $\underline{ 0.672}$ |
> > | (c) PIBD | √ | √ | **0.736(0.072)** | **0.667(0.061)** | **0.768(0.124)** | **0.640(0.039)** | **0.684(0.035)** | **0.699** |
> >
> > ---
> > > **Q3) Using CTransPath for TCGA evaluation. Evaluating PIBD with a ResNet-50 encoder (ImageNet transfer) or a non-TCGA-pretrained encoder would strength the study and its findings.**
> > >
> >
> > Thank you for your valuable feedback! We appreciate your understanding of our decision to use a pretrained encoder for pathology. For your concern about using CTransPath, we have added additional experiments with a ResNet50 encoder pretrained only on ImageNet. Also since the limitation of computing resources and time, we temporarily evaluated these experiments on three datasets. We selected the well-performing TransMIL in the unimodal group and the SurvPath in the multimodal group as the comparison methods, as shown in the table below.
> >
> > | Method | BLCA | COADREAD | STAD | Overall |
> > | --- | --- | --- | --- | --- |
> > | TransMIL | $\underline{ 0.588(0.072) }$| 0.699(0.170) | 0.592(0.095) | 0.626 |
> > | SurvPath | 0.570(0.052) | $\underline{ 0.749(0.100)}$ | $\underline{ 0.624(0.057)}$ | $\underline{ 0.648 }$|
> > | PIBD | **0.595(0.061)** | **0.784(0.124)** | **0.644(0.063)** | **0.674** |
> >
> > The results clearly demonstrate that even when employing encoders not pretrained on TCGA data, PIPD exhibits great improvement compared to the comparison methods, which further strengthens the superiority of our method. We will conduct more comprehensive comparative experiments in the future and incorporate them into subsequent versions.
> >
> > ---
> > > **Q4) How are CLAM-SB / CLAM-MB adapted for survival analysis?**
> > >
> >
> > Yes, the CLAM framework was originally designed for slide classification. To adapt CLAM-SB and CLAM-MB for survival analysis, we followed the setting of comparative experiments conducted in the works [1,2]. Since NLL loss used for survival prediction is a cross-entropy-like loss, CLAM is easily adapted to this loss by modifying the classifier head.
> >
> > **REFERENCES**
> >
> > [1] Xu Y, Chen H. Multimodal Optimal Transport-based Co-Attention Transformer with Global Structure Consistency for Survival Prediction[C]//Proceedings of the IEEE/CVF International Conference on Computer Vision (ICCV), 2023, pp. 21241-21251
> >
> > [2] Zhou F, Chen H. Cross-Modal Translation and Alignment for Survival Analysis[C]//Proceedings of the IEEE/CVF International Conference on Computer Vision. 2023: 21485-21494.

---

> ### Comment · Reviewer_ottr · 2023-12-04
> **To the authors [1/2]**
>
> To the authors,
>
> Thank you for your detailed comment in addressing the points raised. From carefully re-reading the updated manuscript, new experiments, and references, here are my comments:
>
> **Response to Q1) Limited findings w.r.t. clinical application / multimodal interpretability**:
>
> **Contribution**: I agree with the authors and with Reviewer G86X that probabilistic approaches for pathology is a relatively under-explored but promising direction. Regarding notions of "modality-specific/shared" and "intra-/inter-modal redundancy" and comparisons to previous works):
> - "Modality-shared information": This claim is well-supported. In the reviewer's assessment, shared information is modeled via aggregating patch-level pathology feature embeddings based on their similarity with genomic pathway embeddings.
> - "Modality-specific information": This claim is more nuanced, and slightly disagree that SurvPath and MOTcat "ignored the distinct value of modality-specific information". In MOTcat, following computing the OT plan from pathology to genomics (similar to co-attention), genomic pathway features are still explicitly introduced in the risk prediction layer. In SurvPath, genomic-to-genomic relationships are still modeled using Transformer attention and fed into the risk prediction layer. The authors should clarify that genomic features (though not explicitly modeled to have independence with pathology) are still used for survival prediction. Pathology features (without genomic guidance) are not used, supporting the authors' claim.
> - "Intra-modal redundancy": I disagree more with this point. Redundancy in pathology-specific modality is performed via co-attention / OT, where pathology embeddings are pooled based on their similarity to genomic pathway embeddings. Using the processed genomics data made available by SurvPath, at most only 331 pathway tokens are used. As pathways are somewhat prototypical and semantic in describing a unique biological process (different from repeating histology patterns for pathology), there is less motivation to mitigate redundancy here as the tokens are relatively semantically-dense and much lower than bag sizes for WSIs.
> - "Inter-modal redundancy": This claim is well-supported.
>
> **Interpretability and Clinical Application**: I still stand by my original point that the attention heatmaps are difficult to interpret and draw conclusions from. To be more specific:
> - "We have visualized similarity scores of instances to each prototype of risk level and presented top-3 relevant patches and biological pathways, as well as top-3 irrelevant biological pathways":
> - - As the technical contributions of PIBD are in mitigating intra/inter-modal redundancy and emphasizing modality-specific/shared features, it would be useful to understand which specific features in pathology and genomics are captured, and which features are shared across pathology and genomics modality. At the moment, the interpretability in Figure 6 and 7 are similar to that of MCAT / SurvPath / MOTcat in visualizing pathology-genomic correspondences despite having a very different approach. What is modality-specific/shared?
> - - What does "Similarity Scores of Pathways" show? Font sizes are too small. Common practices for reporting pathology images such as magnification per pixel are missing. Variables such as $t$ and $c$ are not defined within the figure.
> - "Since prototypes represent the distribution for each risk band, they inherently possess a global nature...":
> - - With number of prototypes $P=8$, what are the these prototypes learned for pathology and genomics? For pathology, do they relate to a distinct or important morphology? For genomics, do they reflect general "tumor-suppressing" or "immune-activating" pathways (which pathways get grouped together)? For multimodal interpretability, which prototypes are shared and which are conserved? In Figure 6 and 7, it is not clear what each prototype is. Figure 4 is also not very informative on which prototypes are learned. From existing prototype-based works in pathology like H2T [1], I believe the authors should be able to perform a more detailed interpretation of prototypes.
>
> Empirically, this method still performs well. My concern here is that the inductive biases that PIBD solves (intra-/inter-modal redundancy) are not properly investigated and validated. As pathology has many domain-specific challenges for learning representaitons but also unique opportunities for proposing new methods that address domain-specific inductive biases, such inductive biases such as intra-/inter-modal redundancy should have more validation to strengthen its contributions in both ICLR and in pathology.

---

> > ### Comment · Reviewer_ottr · 2023-12-04
> > **To the authors [2/2]**
> >
> > **Response to Q2) Ablation Experiments for PIB/PID. Ablation experiments for PID/PIB can have stronger baselines.** I thank the authors for their effort in improving the baselines, which addressed my concerns
> >
> > **Response to Q3) Using CTransPath for TCGA evaluation.** I thank the authors for their effort in adding additional experimentation with ResNet-50 features. Ideally, the reviewer would like to see more investigation with all cancer types and more methods, but think that the comparison of methods are fair and does not diminish the method contributions. This point should also be included in the limitations of this work.
> >
> > **Response to Q4) How are CLAM-SB / CLAM-MB adapted for survival analysis?** Author's point has addressed my concerns.

---

### Official Review · Reviewer_HNZs · 2023-11-01

**Soundness:** 3 good
**Presentation:** 3 good
**Contribution:** 3 good
**Rating:** 8
**Confidence:** 5

**Summary:**

The authors introduce a novel framework, Prototypical Information Bottlenecking and Disentangling (PIBD), comprising a Prototypical Information Bottleneck (PIB) module for handling intra-modal redundancy and a Prototypical Information Disentanglement (PID) module for addressing inter-modal redundancy. Extensive experiments conducted on five cancer benchmark datasets establish their method's superiority over other approaches.

**Strengths:**

(1) The concept of Information Bottleneck (IB) is intriguing, as it offers a promising solution to eliminate unnecessary redundancy, and the PIB module effectively addresses computational challenges.
(2) The technical details are presented with clarity and precision.
(3) The thoroughness of the experiments conducted to validate the effectiveness of each component is commendable.
(4) The quality of the visual presentations is noteworthy.

**Weaknesses:**

(1) The Prototypical Information Disentanglement (PID) module could benefit from improved clarity in its description.

**Questions:**

(1)What is the significance of the red arrows in Figure 1, particularly within the context of the PID module?
(2)How does PIB save the computation? Please make quantitative analysis.

---

> ### Author Response · Authors · 2023-11-17
> **To Reviewer HNZs**
>
> Thank you for your encouraging words and constructive comments. We sincerely appreciate your time in reading the paper. In the following, your comments are first stated and then followed by our point-by-point responses.
>
> ---
> > **Q1) The Prototypical Information Disentanglement (PID) module could benefit from improved clarity in its description.**
> >
>
> Regarding the clarity of the PID module, we have revised the corresponding **section 3.3 on Page 6** in the manuscript to enhance the understanding for readers.
>
> ---
> > **Q2) What is the significance of the red arrows in Figure 1, particularly within the context of the PID module?**
> >
>
> The red arrows in Figure 1 signify the computation of the joint prototypical distribution, where each arrow connects to the positive prototype of a sample. The calculated joint distribution gives guidance for extracting modality-common and modality-specific information.
>
> ---
> > **Q3) How does PIB save the computation? Please make quantitative analysis.**
> >
>
> Thank you for your feedback. For the computation efficiency of PIB, it can be known that directly employing VIB would lead to intractable bag-level distribution $p(z|\bf{x})$ due to the large number of individual instance distributions within a bag $\bf{x}$, based on the analysis in the second paragraph **on Page 5**.  The proposed PIB can simplify this issue by modeling bag-level distribution with prototypes.
>
> We can coarsely quantify the reduction in computational complexity compared to the original VIB. Assuming the sampling frequency is $K$, the number of samples per iteration is $N$, and there are $M$ instances of $D$-dimension in a bag, the computational complexity is $O(K\cdot N\cdot M\cdot D)$. However, in PIB, we only need to sample from prototypes. Assume that the number of prototypes is $P$, the complexity becomes  $O(K\cdot N\cdot P\cdot D)$, where $P << M$. For example, P is 8 in our work, while M is usually larger than 10,000.

---

### Official Review · Reviewer_bKh7 · 2023-11-01

**Soundness:** 4 excellent
**Presentation:** 4 excellent
**Contribution:** 4 excellent
**Rating:** 10
**Confidence:** 4

**Summary:**

This paper introduces a novel approach to predicting survival outcome by combining histology and genomics data. The model incorporates two novel components based on information bottlenecks in information theory: prototypical information bottleneck and prototypical information disentanglement.

**Strengths:**

The problem is well defined and the introduced method is appropriate with detailed explanation and evaluation. Results represent a meaningful improvement on prior state of the art approaches. The analysis is thorough and builds upon established practices in the literature. Although the advances over the state of the art are modest, it does represent a consistent improvement compared to other methods across all datasets.

**Weaknesses:**

There are very few weaknesses with this paper.

The evaluation could be improved by adding a “naive” combination of the risk scores from genomics and histology. This could be performed by taking the best performing models which look at an individual modality – i.e. SNNTrans and CLAM-MB – and then combining the predictions of these models using a Cox proportional hazards model against the original survival outcome data. This would act as a suitable baseline to see to what extent learning the combined risk prediction model as described in this work outperforms a naive approach to combining the risk prediction across the two modalities. This would allow the reader to fully understand the added benefit of the model described over the other approaches.

**Questions:**

Kaplan-Meier analysis: please state how cut-offs were selected in the manuscript.

Please see comment about combining individual modality predictions using CoxPH as a “naive” baseline.

The appendix gives some discussion of hyperparameter choices, although it is not clear how these were selected with the dataset. As cross validation is used to generate the evaluation results, to what extent were the hyperparameters selected using the evaluation datasets and is there risk of selection bias as a result?

---

> ### Author Response · Authors · 2023-11-17
> **To Reviewer bKh7**
>
> Thank you for your encouraging words and constructive comments. We sincerely appreciate your time in reading the paper. In the following, your comments are first stated and then followed by our point-by-point responses.
>
> ---
>
> > **Q1) A “naive” baseline. The evaluation could be improved by adding a “naive” combination of the risk scores from genomics and histology. i.e. Combining individual modality predictions using CoxPH.**
> >
>
> Thank you very much for this insightful comment! We follow your suggestion and combine the predictions of SNNTrans and CLAM-MB as a “naive” baseline. The results are added in **Table 1, Page 8**, and are also shown in the table below.
>
> | Methods | BRCA | BLCA | COADREAD | HNSC | STAD | Overall |
> | --- | --- | --- | --- | --- | --- | --- |
> | SNNTrans | 0.679(0.053) | 0.583(0.060) | 0.739(0.124) | 0.570(0.035) | 0.547(0.041) | 0.622 |
> | CLAM-MB | 0.696(0.098) | 0.623(0.045) | 0.721(0.159) | 0.620(0.034) | 0.648(0.050) | 0.662 |
> | SNNTrans+CLAM-MB(naive combination) | 0.699(0.064) | 0.625(0.060) | 0.716(0.160) | 0.638(0.066) | 0.629(0.065) | 0.661 |
> | PIBD | **0.736(0.072)** | **0.667(0.061)** | **0.768(0.124)** | **0.640(0.039)** | **0.684(0.035)** | **0.699** |
>
> From the results, it's clear that the "naive" combination approach at the prediction level yielded unsatisfactory results. The main reason behind this suboptimal performance is that this method focuses solely on single-modal learning, failing to extract relevant common information between modalities. This further underscores the superiority of our approach, which, through information disentanglement, eliminates redundancy between inter-models and effectively captures both specific and common information.
>
> ---
> > **Q2) Kaplan-Meier analysis. Please state how cut-offs were selected in the manuscript.**
> >
>
> Thank you for your suggestion.  We have provided more details about the Kaplan-Meier analysis in our manuscript. In our work, we employed a five-fold cross-validation evaluation, getting one model for every data fold. For every validation set, we determined the cut-offs by taking the median of the model-predicted risk values. Patients with a risk higher than the median are assigned as high risk, and patients with a risk lower than the median as low risk. In addition, in response to the suggestion from reviewer G86X, we have also reported the median survival months for each risk group on **Page 8**.
>
> ---
> > **Q3) The hyperparameter choices. How hyperparameters were selected with the dataset. As cross-validation is used to generate the evaluation results, to what extent were the hyperparameters selected using the evaluation datasets and is there risk of selection bias as a result?**
> >
>
> Thank you for bringing up this important point.  Due to the limited size of datasets, like many other methods, we chose cross-validation as the assessment approach, allowing us to effectively leverage the available data and enhance the reliability of our evaluation.
>
> In our study, we adopted a careful approach to address the potential risk of selection bias in hyperparameter choices. To begin with, we defined the hyperparameter search space based on the commonly used ranges in information bottleneck methods like VIB[1] and MIB[2]. During the selection process, we implemented a grid search strategy. For each hyperparameter, while keeping others fixed, we conducted a five-fold cross-validation, selecting the hyperparameter that exhibited the best average performance across all validation sets. Furthermore, to further mitigate selection bias, these hyperparameters were selected on multiple datasets, ensuring the generalizability of the chosen configuration. By employing this approach, we aimed to minimize the risk of selection bias.
>
> we have incorporated the introductions on this aspect into **Appendix C.2 on Page 19**.
>
> **REFERENCES**
>
> [1]  Alemi, Alexander A., et al. "Deep Variational Information Bottleneck." *International Conference on Learning Representations*. 2016.
>
> [2] Federici, Marco, et al. "Learning Robust Representations via Multi-View Information Bottleneck." *8th International Conference on Learning Representations*. OpenReview. net, 2020.

---

### Author Response · Authors · 2023-11-21

Dear reviewers,

We wish to convey our sincere gratitude once again for the invaluable time and effort you have dedicated to reviewing our submission.

As we approach the end of the rebuttal period, we kindly inquire whether our responses have effectively addressed your concerns or questions. We are committed to addressing any remaining concerns with utmost eagerness.

We sincerely look forward to your responses.

Respectfully,

Authors

---

### Meta-Review · Area_Chair_vsEM · 2023-12-05

**Metareview:**

This paper received the following ratings after rebuttal: 10, 8, 5, 6.
So, the majority of the reviews are leaning towards acceptance, and just one score is below threshold.
The authors wrote a careful reply to all reviewers' remarks, and this led to clear positive scores from 2 reviewers. Indeed, another reviewer gave originally a higher score, which was lowered during the last discussion with the most critical reviewer.

Overall, the proposed approach was deemed sufficiently original, technically sound, with a comprehensive experimental validation and ablation studies.
The utility for domain scholars was also stressed, even if the "interpretability" of the results was still considered an issue despite not so significant to justify a rejection.

For these reasons, this paper can be accepted for publication at ICLR 2024.
In the final version, authors are recommended to better explain the interpretability of the results, which are not so well illustrated and commented, in particular Fig. 6 and 7 in the appendix are far from being clearly explained.

**Justification For Why Not Higher Score:**

Not unanimous positive evaluations

**Justification For Why Not Lower Score:**

2/3 reviewers out of 4 were quite positive, the work seems interesting to be raised more to the attention of the audience.

---

### Decision · Program_Chairs · 2024-01-16

Accept (spotlight)